# Activating Parkin-dependent mitophagy alleviates oxidative stress, apoptosis, and promotes random-pattern skin flaps survival

Zhengtai Chen[1,2,5], Hongqiang Wu[1,2,5], Jianxin Yang[1,2], Baolong Li[1,2], Jian Ding[1,2], Sheng Cheng[1,2], Nageeb Bsoul[1,3], Chenxi Zhang[4], jiaorong li[3], Haixiao Liu [1,2✉], Damu Lin [1✉] & Weiyang Gao [1,2✉]

The random-pattern skin flap is a crucial technique in reconstructive surgery and flap necrosis caused by ischemia/reperfusion injury is a major postoperative complication. Herein, we investigated the mechanism of mitophagy induced by Melatonin (ML) and its effect on the survival of skin flaps. Our results demonstrated that ML could activate mitophagy, ameliorate oxidative stress and alleviate apoptosis in Tert-Butyl hydroperoxide solution (TBHP)-stimulated human umbilical vein endothelial cells in vitro. Inhibiting ML-induced mitophagy considerably abolished its protective effects. Moreover, knockdown of Parkin by siRNA inhibited ML-induced mitophagy, and subsequently exacerbated oxidative stress and apoptosis. Further study demonstrated that inhibition of AMPK reversed these protective effects of ML and downregulated the expression of TFEB. In the vivo study, ML effectively promoted flap survival by activating mitophagy and subsequently ameliorating oxidative stress and mitigating apoptosis. These results established that ML is a potent agent capable for increasing random-pattern skin flap survival by activating Parkin-dependent mitophagy through the AMPK-TFEB signaling pathway.

---

[1] Department of Orthopedics, The Second Affiliated Hospital and Yuying Children's Hospital of Wenzhou Medical University, Wenzhou 325000 Zhejiang Province, China. [2] Zhejiang Provincial Key Laboratory of Orthopedics, Wenzhou 325000 Zhejiang Province, China. [3] Wenzhou Medical University, Wenzhou, China. [4] Department of Orthopedics, The Number 6 Hospital of Ningbo, Ningbo, China. [5]These authors contributed equally: Zhengtai Chen, Hongqiang Wu. ✉email: spineliu@163.com; ensita@163.com; weiyanggaoi@126.com

Skin flaps transfer is an essential technique for reconstructive surgery. Random-pattern skin flaps without an axial vasculature are among the most commonly used flaps in wound repair and plastic surgeries due to the freedom of position and arc of rotation[1,2]. However, the random pattern skin flap is prone to necrosis at the distal end of the flap precisely due to the lack of a specific arteriovenous system and blood supply and thereby limiting flap size. Earlier studies have reported that insufficient blood supply and subsequent ischemia-reperfusion (I/R) injury is the primary cause of flap necrosis[3,4]. During the process of I/R, oxidative stress injury plays a critical factor. Following oxidative stress, massive accumulation of reactive oxygen species (ROS) can respond non-specifically and promptly with cellular biomolecules, including DNA and proteins which eventually trigger DNA variations, protein oxidation, and lipid peroxidation[5]. Given the mechanisms of I/R injury, strategies to alleviate oxidative stress and reduce cell death have been under active investigation in recent years.

The mitochondrion is the power plant of cells that produce Adenosine triphosphate (ATP) through oxidative phosphorylation, and the electron transport chain is the vital process[6,7]. During energy production in the electron transport chain, ROS are also generated[8,9]. When the extracellular environment alters, mitochondria damaged and the electron transport chain interrupted, leading to the release of a large number of ROS and the disruption of intracellular homeostasis. Mitophagy is selective macroautophagy that can remove damaged mitochondria and avoid a destructive burst of ROS[10]. It was reported that mitophagy exerted protective functions in various pathological processes of several diseases, including spinal cord ischemia-reperfusion injury[11], acute kidney injury[12], and oxidative stress-induced intestinal barrier injury[13]. Thus, activating mitophagy might be a powerful approach to reduce oxidative stress injury during flap translation.

Melatonin (ML) is a time-keeping neurohormone chiefly secreted by the pineal gland[14,15]. In addition to regulating the circadian rhythm, modulating the activity of antioxidants and anti-inflammatory effects also prominent role of melatonin. Recently, the regulatory effect of ML on mitophagy has attracted widespread attention. ML treatment alleviates myocardial MI/R injury by regulating mitophagy through targeting SIRT6[16]. ML against the senescence of mesenchymal stem cells (MSCs), avoids the decrease in the therapeutic effect of MSCs by regulating mitophagy[17]. Previous studies have determined that melatonin improved skin flap viability, but they focused only on the role of antioxidant stress[18,19]. Therefore, our group postulated that ML-induced mitophagy might also be instrumental in flap survival. The main purpose of this study was to further clarify the effect of ML on the survival of skin flaps and the mechanism by which ML regulates mitophagy.

## Results

### The expression of Parkin was increased in rat's flap model and TBHP treated HUVECs.

To determine whether mitophagy participates during flap transfer, we excised tissue from the center part of Area II from sham and control groups (Fig. 1a). Then, the mitophagy-related proteins levels were assessed. As illustrated in Fig. 1b, c, the expression level of Parkin was increased in the control group. Then, we detected two critical autophagy-related proteins LC3II (a marker of autophagosome formation) and p62 (the autophagy cargo receptor). Interestingly, both LC3II and P62 were increased in the control group compared with the sham group (Fig. 1b, c). Afterward, treatment of HUVECs with TBHP was applied to mimic the flap model in vitro. CCK8 results determined that TBHP inhibited HUVECs viability in a dose-dependent manner (Supplementary Fig. 1a). According to the CCK8 results, TBHP (100 μM) was used in subsequent studies. Hence, we detected that TBHP successfully upregulated the expression levels of Parkin, LC3, and P62 (Fig. 1d, e). Furthermore, western blot results revealed that the apoptosis-related protein levels of Bax and C-caspase3 were higher in the TBHP treatment group (Fig. 1f, g). In contrast, the level of the anti-apoptotic protein, Bcl-2, was lower in the TBHP treatment group than in the control group (Fig. 1f, g). TUNEL staining also verified the results of western blot and confirmed the apoptosis level was higher in the TBHP group compared with control group (Fig. 1h, i). In addition, the oxidative level was assessed by western blot, and results revealed that TBHP restrained the expression levels of anti-oxidative stress-related proteins (Fig. 1k, l). Besides, MitoSox staining revealed an increased level of mitochondrial ROS in the TBHP group causing mitochondrial dysfunction (Fig. 1j, m). Taken together, TBHP-exposed HUVECs exhibited mitochondrial impairment, accumulation of ROS, and an increased level of apoptosis.

### ML promoted mitophagy and ameliorated TBHP-induced apoptosis in HUVECs.

First, CCK8 results showed that the ML concentrations within 100 μM had no toxicity to HUVECs (Supplementary Fig. 1b). Thus, ML (100 μM) was used in subsequent studies. To evaluate mitophagy activation, UA, as one of the best-characterized mitophagy inducers so far, was used as the positive control in this experiment[20]. Afterward, we detected the expression of LC3II and p62 and results exhibited that the level of LC3II was increased after TBHP treatment and was considerably higher in TBHP + ML and TBHP + UA groups compared with the TBHP group (Fig. 2a, b). The expression of P62 was upregulated in the TBHP group but downregulated in both TBHP + ML and TBHP + UA groups (Fig. 2a, b). The formation of mitochondrial autophagosomes is then reflected by the co-localization of mitochondrial outer membrane marker Tom20 and LC3. The co-localization rate of LC3 and Tom20 was found to be higher in the TBHP group than control group (Fig. 2c, d). Although the co-localization rate in the TBHP + ML group was lower than that of the TBHP + UA group, it was considerably higher than that of the TBHP group. This implied that ML effectively promoted the formation of mitochondrial autophagosomes. It is worth noting that autophagy is a dynamic process in which the fusion of autophagosomes and lysosomes is often used as an indicator of the patency of autophagy. Therefore, the increase in the formation of mitochondrial autophagosomes cannot fully prove the activation effect of ML on mitophagy. In this case, a mitophagy detection kit was performed to observe mitophagy. Mtphagy dye is localized to mitochondria and emits fluorescence as it is acidified during autophagy. As depicted in Fig. 2e, f, TBHP did increase the formation of autophagosomes but could not promote the combination of autophagosomes and lysosomes. Conversely, UA and ML not only elevated the number of autophagosomes but also promoted the fusion of mitochondrial autophagosomes and lysosomes. Furthermore, bafilomycin A1 (an inhibitor of lysosomal function) was used to validate the induction of mitophagy flux. As displayed in Fig. 2g, h, the difference in LC3-II and P62 expression levels demonstrated that mitophagy flux was considerably enhanced by ML. Furthermore, western blot results demonstrated that the level of Bax and C-caspase3 was lower in TBHP + ML and TBHP + UA groups compared with the TBHP group (Supplementary Fig. 2a, b). In contrast, Bcl-2 was upregulated in TBHP + ML and TBHP + UA groups compared with the TBHP group.

To further investigate the effect of ML on mitochondria, we used a MitoTracker dye and transmission electron microscopy to

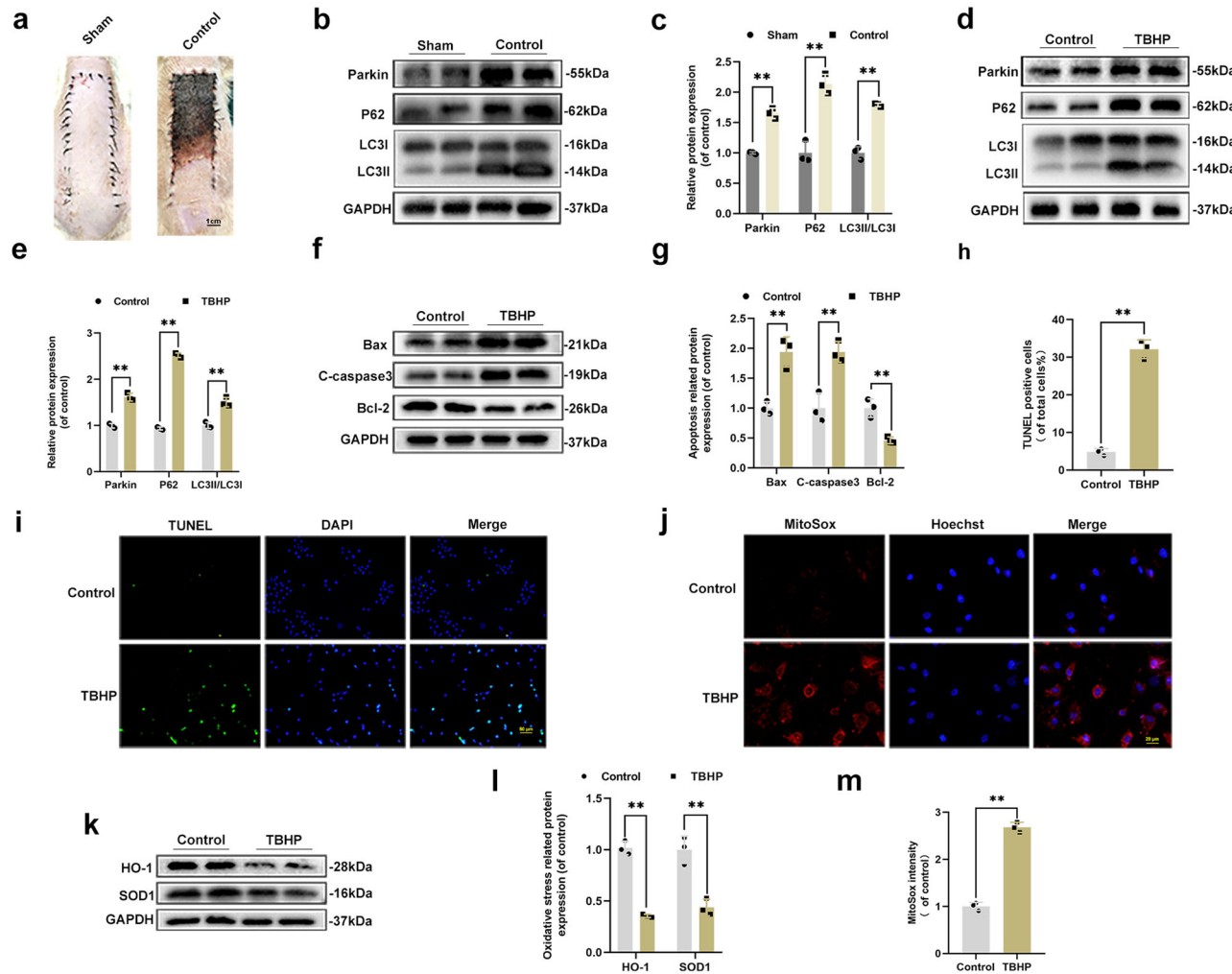

**Fig. 1 The expression of Parkin was increased in rat's flap model and TBHP-treated HUVECs. a** Digital photograph of the flap in sham and control groups on POD 7 (bar: 1 cm). **b**, **c** The protein expression level of Parkin, LC3, and P62 in sham and control groups. **d**, **e** The protein expression level of Parkin, LC3, and P62 of HUVECs in control and TBHP treatment groups. **f**, **g** The protein expression level of Bax, C-caspase3, and Bcl-2 in HUVECs in the control and TBHP treatment groups. **h**, **i** TUNEL staining was used to detect apoptosis in HUVECs after TBHP stimulation (bar: 50 μm). **k**, **l** Western blot showing the protein expression of HO-1 and SOD1 in control and TBHP treatment groups. **j**, **m** MitoSox staining was performed to detect the mitochondrial ROS in HUVECs (bar: 20 μm) and the fluorescence intensity was quantified. All experiments were repeated in triplicates ($n = 3$), and the data are presented as mean ± S.D. $n = 3$, **$P < 0.01$, *$P < 0.05$.

measure morphology of mitochondria in the presence and absence of bafilomycin A1. MitoTracker fluorescence images showed that ML treatment prevented MitoTracker intensity from decreasing, whereas bafilomycin A1 reversed this effect (Supplementary Fig. 2c, d). The transmission electron microscopy images revealed that while the number of mitochondria in the ML + TBHP group was lower than in the THBP group, the mitochondrial morphology was more complete, with a sharper mitochondrial crest, intact mitochondrial membrane, and greater autophagolysosome formation. However, BafA1 blocked these effects of ML. In the THBP + ML + BafA1 group demonstrated mitochondrial vacuolation, shrinkage and mitochondrial crest fracture (Supplementary Fig. 2e). To sum up, our results suggest that ML activated mitophagy and ameliorated TBHP-induced apoptosis in HUVECs.

**ML protected HUVECs from TBHP-induced injury in a mitophagy-dependent manner.** CsA, a mitophagy inhibitor, downregulated the protein levels of LC3, Parkin, and PINK1, while upregulating P62, indicating that ML-induced mitophagy

was successfully blocked, as shown in Fig. 3a, b. Then, we investigated apoptosis and oxidative stress-related proteins and found that the expression levels of C-caspase3 and Bax increased, whereas Bcl-2, HO-1, and SOD1 were down-regulated which suggested that the protective effect of ML was abolished by CsA (Fig. 3c, d). Examination of MMP by JC-1 staining revealed that MMP was decreased in the TBHP group but restored after co-administrated with ML. Furthermore, mitophagy deficiency could reverse MMP induced by ML and result in mitochondrial injury (Fig. 3e, f). For the assessment of mitochondrial metabolism, ATP production analysis suggested that ML enhanced mitochondria-derived ATP, while CsA blocked ML's effect on mitochondria energy production (Fig. 3g). Moreover, to further provide evidence regarding the mechanism of action of ML on mitophagy, another mitophagy inhibitor, Mdivi-1, was used. Consistent with the results above, Mdivi-1 successfully inhibited ML-induced mitophagy and largely abolished the anti-oxidative stress and anti-apoptotic effects of ML (Supplementary Fig. 3a–d). Altogether, ML-induced mitophagy played a key role in ameliorating oxidative stress and apoptosis in HUVECs.

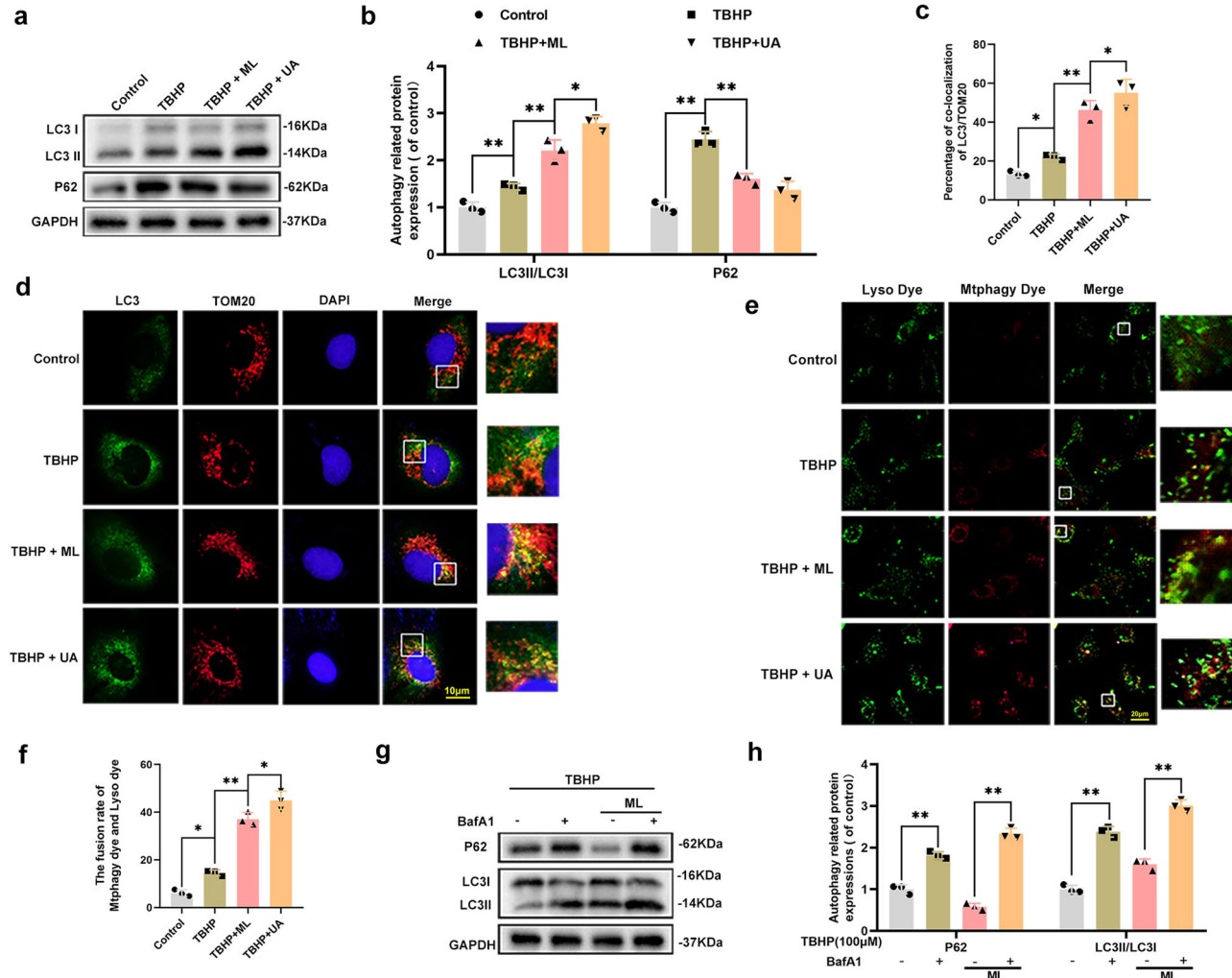

**Fig. 2 ML promoted mitophagy and ameliorated TBHP-induced apoptosis in HUVECs. a**, **b** Western blot showing the protein expression level of LC3 and P62 in HUVECs in the control, TBHP, TBHP + ML, and TBHP + UA groups. **c**, **d** Image of double-labeled staining of Tom20 and LC3 in HUVECs in the above groups (Green: LC3, red: TOM20, bar: 10 μm). **e**, **f** Mitophagy detection kit detected the mitophagy in HUVECs in the above groups, wherein Mtphagy Dye staining indicates mitophagy, Lyso Dye staining indicates lysosomes, and yellow staining indicates co-localization of mitophagy and lysosomes. Scale bar: 20 μm. **g**, **h** The expression level of LC3 and P62 in HUVECs in the presence and absence of Bafilomycin A1. All experiments were performed at least 3 times ($n \geq 3$). Data are presented as mean ± S.D. $n = 3$, **$P < 0.01$, *$P < 0.05$.

**ML promoted mitophagy by upregulating Parkin**. Three classical pathways are fundamental in mitophagy, among which the PINK1/Parkin pathway is the most widely studied[21–23]. To study the biological manner of HUVECs after TBHP and ML treatment, samples from each group were sequenced. A total of 2085 differentially expressed genes were identified, including 1186 upregulated genes and 899 downregulated genes in the TBHP group and the TBHP + ML groups (Fig. 4a). The heatmap of the three groups illustrated that the mitophagy-related genes PRKN(Parkin) and PINK1 were upregulated in the TBHP group and were even higher in the TBHP + ML group (Fig. 4b). In light of the western blot and sequencing results, we considered whether ML-induced mitophagy was activated in a Parkin-dependent manner. Therefore, we utilized siRNA to knock down the Parkin gene. As presented in Fig. 4c, d, Parkin was successfully downregulated by the siRNA. Afterward, the western blot results of the levels of LC3 and p62 demonstrated that Parkin knock-down effectively inhibited autophagy (Fig. 4e, f). Besides, the decline of MitoTracker fluorescence intensity was found in the Parkin knock-down group (Supplementary Fig. 4a, b). Furthermore, the sequencing results determined that ML treatment reduced

apoptosis which was reflected by the decreased level of two critical apoptosis-related proteins Bax (Bak) and cytochrome C (CYC, one of the main proteins that link mitochondrial damage and the caspase family) in the TBHP + ML group, compared to the TBHP group (Fig. 4a, b). Additionally, the western blot results showed that the anti-apoptotic effects of ML were blocked by Parkin inhibition (Fig. 4g, h). To measure levels of oxidative stress, SOD1 and HO-1 levels were found to decrease after Parkin knockdown, which implied that Parkin depletion remarkably inhibited ML's anti-oxidative stress effects in TBHP-treated HUVECs (Fig. 4g, h). The MitoSox assay monitored the ROS formation in mitochondria, and it was observed that Parkin depletion substantially induced ROS production (Fig. 4i, k). Moreover, a TEM and ATP detection kit were used to evaluate the effect of Parkin knockdown on mitochondrial damage. TEM images exhibited the mitochondrial damage including mitochondrial vacuolation, shrinkage, rupture, and fewer mitophagosomes in the si-Parkin group, which indicated that Parkin inhibition largely abolished the effects of ML (Fig. 4j). Similarly, ML treatment successfully restored ATP synthesis which decreased by TBHP, whereas Parkin knockdown effectively

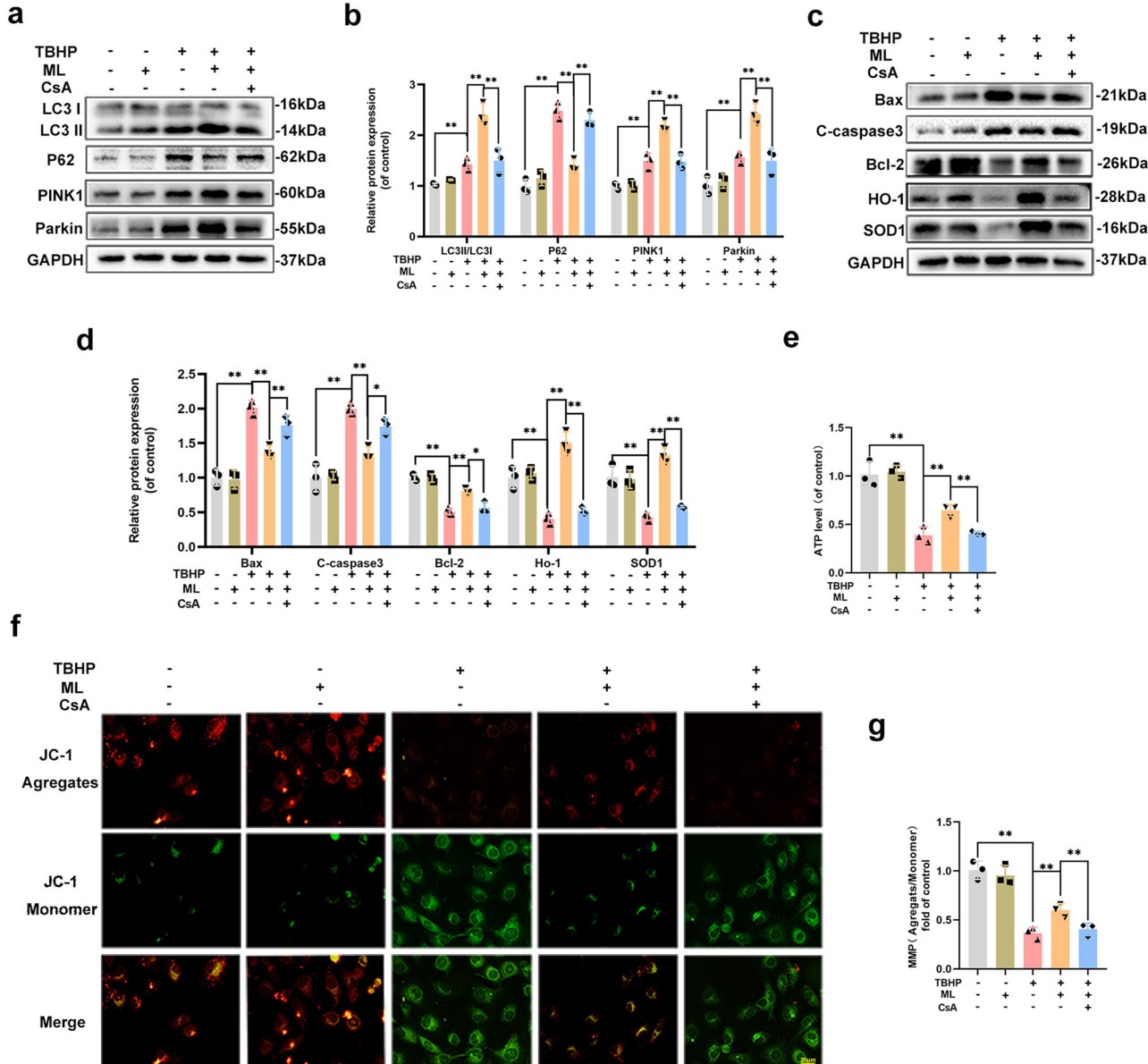

**Fig. 3 ML protected HUVECs from TBHP induced injury in a mitophagy-dependent manner. a, b** The protein expression level of LC3-II, p62, Pink1, and Parkin in HUVECs was quantified and showed by a histogram. **c, d** Expression and quantification of apoptosis-related and oxidative-related proteins Bax, C-caspase3, Bcl-2, HO-1, and SOD1 were measured by western blot. **e, f** JC-1 probe was performed to detect the change of MMP and was quantified and showed by a histogram (bar: 20 μm). **g** ATP production in HUVECs was measured by an ATP detection kit and showed by a histogram. All experiments were performed at least 3 times ($n \geq 3$). Data are presented as mean ± S.D. $n = 3$, **$P < 0.01$, *$P < 0.05$.

decreased ATP production (Fig. 4l). Collectively, these data suggest that Parkin is essential for ML-induced mitophagy.

**ML induced mitophagy via upregulating the nuclear translocation of TFEB in HUVECs.** TFEB is known for its ability to regulate lysosomal biogenesis and autophagy[24,25]. Whether TFEB participates in ML-induced mitophagy is unknown. According to the RNA sequencing results, the TFEB gene was downregulated by TBHP treatment and upregulated after ML treatment, signifying that TFEB participated in ML-induced mitophagy (Fig. 4b). As seen in the western blot results, TBHP suppressed TFEB nuclear expression whereas ML increased it, as compared to the control group (Fig. 5a, b). It's worth mentioning that the cytoplasmic TFEB expression level was somewhat higher in the ML group, although this was not statistically significant. Similarly, TFEB immunofluorescence staining validated the western blot

results that ML promotes the nuclear translocation of TFEB (Fig. 5c, d). To confirm the role of TFEB in ML-induced mitophagy, a TFEB si-RNA was used. The western blot revealed that si-TFEB effectively suppressed TFEB (Fig. 5e, f). Then the autophagy-related proteins LC3 and P62 were next measured. After TFEB knockdown, the decreased expression of LC3 and increased expression of P62, indicating that ML-induced autophagy was inhibited (Fig. 5g, h). Additionally, the upregulated C-caspase3 and downregulated SOD1 expression levels, reflected the anti-apoptotic and anti-oxidative effects of ML were antagonized by TFEB knockdown (Fig. 5g, h). All in all, it could be speculated that ML-induced mitophagy in HUVECs may through upregulating the nuclear translocation of TFEB.

**ML induced mitophagy via AMPK-TFEB signaling pathway in HUVECs.** Previous studies reported that AMPK plays an

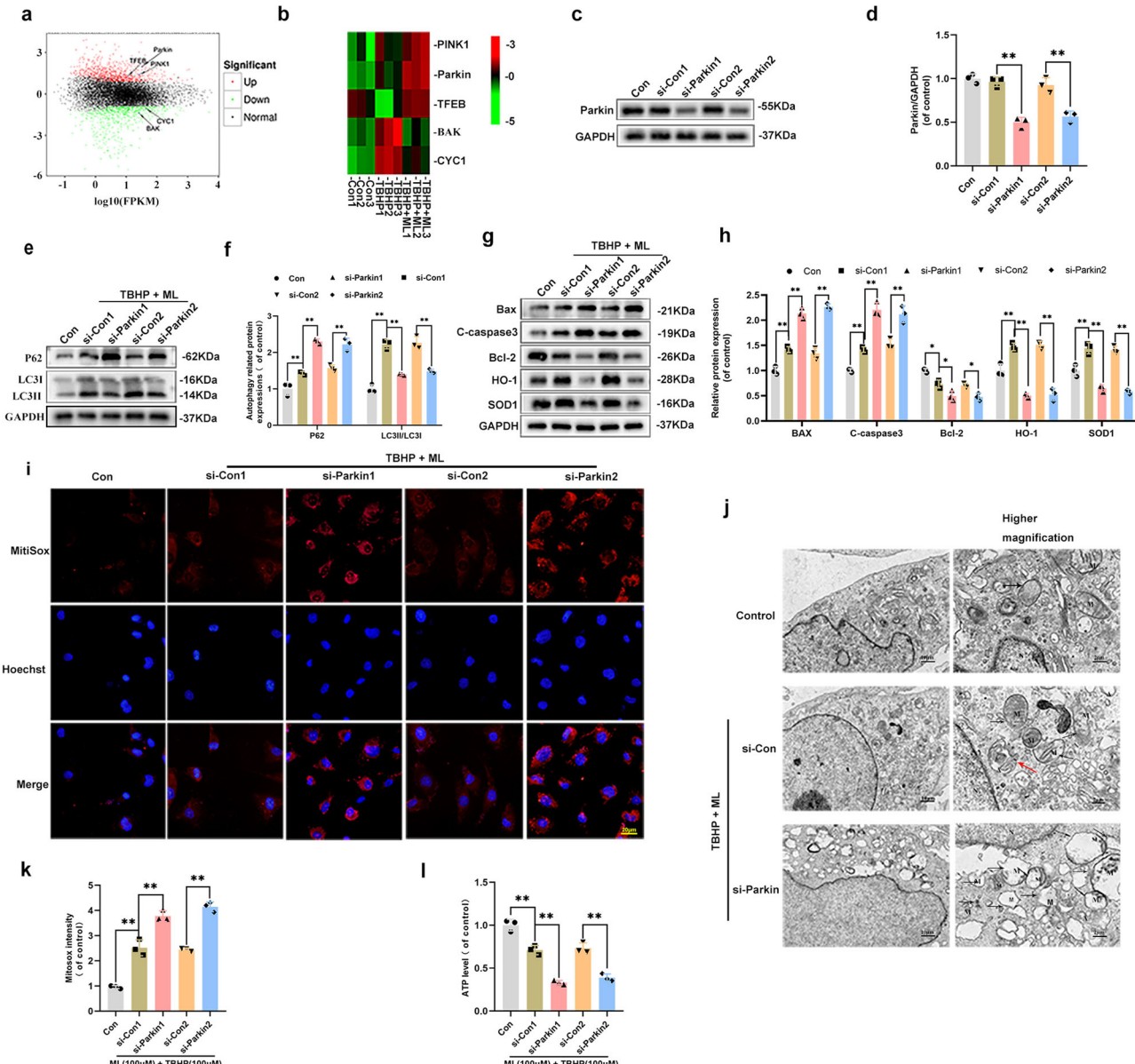

**Fig. 4 ML promoted mitophagy by upregulating Parkin. a** M-versus-A (MA) plot was shown the differential gene expression between TBHP and TBHP + ML groups. **b** Heatmap analyses of the key molecules differentially regulated in HUVECs in control, TBHP, and TBHP + ML groups based on transcriptome analysis. **c, d** The expression of Parkin was evaluated by western blots in HUVECs treated with si-RNA. **e, f** Western blot measured the expression level of LC3 and P62 after Parkin knockdown. **g, h** Expression and quantification of protein Bax, C-caspase3, Bcl-2, HO-1, and SOD1 were measured by western blot. **i** MitoSox staining was performed to detect the mitochondria ROS in HUVECs (bar: 20 µm). **j** Detection of ultrastructure of mitochondria and autophagic change by TEM (×10,000 or 50,000) (Black arrow: swollen mitochondria with fractured cristae; Red arrow: autophagolysosome; M: mitochondria; A: autophagosome). **k** The fluorescence intensity was quantified and demonstrated by a histogram. **l** ATP production in HUVECs was measured by an ATP detection kit and showed by a histogram. All experiments have been performed at least 3 times ($n \geq 3$). Data are presented as mean ± S.D. $n = 3$, **$P < 0.01$, *$P < 0.05$.

essential role in regulating TFEB transcription[13,26]. To explore the role of the AMPK-TFEB signaling pathway in ML-induced mitophagy, Compound C (CC), an AMPK inhibitor, was used. Western blot results demonstrated that after ML administrated, the level of phosphorylation of AMPK was increased but declined when co-administrated with CC. Parkin and Nuclear TFEB expressions exhibited a similar trend with p- AMPK in ML + TBHP and ML + TBHP + CC groups (Fig. 6a–d). However, there was no significant difference in cytoplasmic TFEB among the four groups. Moreover, co-localization staining of Parkin and Tom20 reflected the mitophagy alternations, when AMPK has inhibited the co-localization rate of Parkin and Tom20 was declined

(Fig. 6e, h). Besides, TUNEL staining revealed an increased incidence of apoptosis in HUVECs of the ML + TBHP + CC group, compared with the ML + TBHP group (Fig. 6f, g). In summary, our results confirmed that ML induced mitophagy in HUVECs mainly through the AMPK-TFEB signaling pathway.

**ML promoted random-pattern skin flap survival and was reversed by the AMPK inhibitor.** According to the previous study and the results of the in vitro experiment, a random pattern skin flap model was constructed to investigate the effects of ML in vivo. On POD 3, the flap became hyperpigmented and stiff in Area III, and the control group had the highest rates of necrosis,

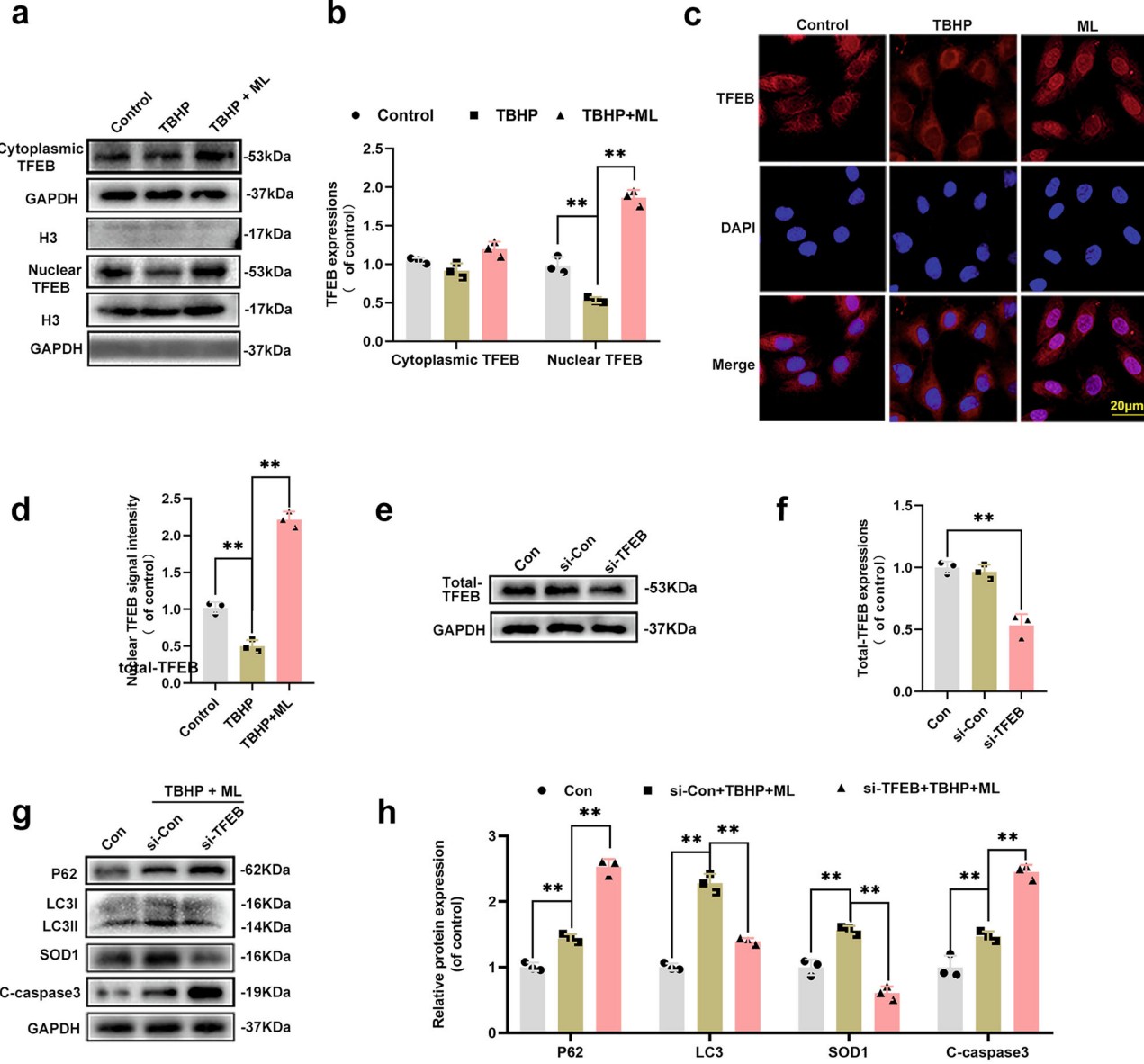

**Fig. 5 ML induced mitophagy via upregulating the nuclear translocation of TFEB in HUVECs. a, b** The expression of cytoplasmic and nuclear TFEB in HUVECs in control, TBHP and TBHP + ML groups. **c, d** Subcellular localization of TFEB measured immunofluorescence in control, TBHP and TBHP + ML groups (bar: 20 μm). **e, f** The protein expression of whole TFEB and in HUVECs was detected by western blot. **g, h** Western blots were used to measure the expression of protein P62, LC3, C-caspase3 and SOD1. All experiments have been performed at least 3 times. Data are presented as mean ± S.D. $n = 3$, **$P < 0.01$, *$P < 0.05$.

but there was no significant difference between the ML and ML + CC groups (Fig. 7a, b). On POD 7, the necrosis area expanded in each group, with the ML group exhibiting dramatically superior survival than the control group, but co-administration of CC reversed the effect of ML (Fig. 7c, d). Laser Doppler was performed to visualize blood supply under the flap. On POD 3, there was no difference among the three groups (Fig. 7e, f). However, on POD 7, the ML group demonstrated the highest blood flow intensity compared to the ML + CC and control groups, while it was lowest in the control group (Fig. 7g, h). As shown in Fig. 7i, j, tissue edema measurement results showed that ML could reduce flap edema, while CC largely abolished this effect. Furthermore, HE staining was performed, and we found that the vessel density was higher in the ML group, while leukocytic infiltration was lower compared to the control group. Besides, there was no major difference between control

and ML + CC groups (Fig. 7k). In summary, our findings prove that ML improves skin flaps survival.

**In vivo, ML reduced apoptosis and oxidative stress via inducing mitophagy.** To investigate the effect of ML in alleviating apoptosis in skin flaps, TUNEL staining was performed. The results showed that the number of TUNEL-positive cells in the ML group was lower than that in the control group but was increased in the ML + CC group (Fig. 8a, b). Afterward, IHC staining was used to measure the expression level of TFEB, Parkin, LC3-II, C-caspase3, and SOD1 in vivo. As shown in Fig. 8c–h, compared with the control group, the expression of TFEB, Parkin, LC3-II, and SOD1 was upregulated in the ML group while co-administration with CC largely antagonized this effect. The expression of C-caspase3 exhibited a completely

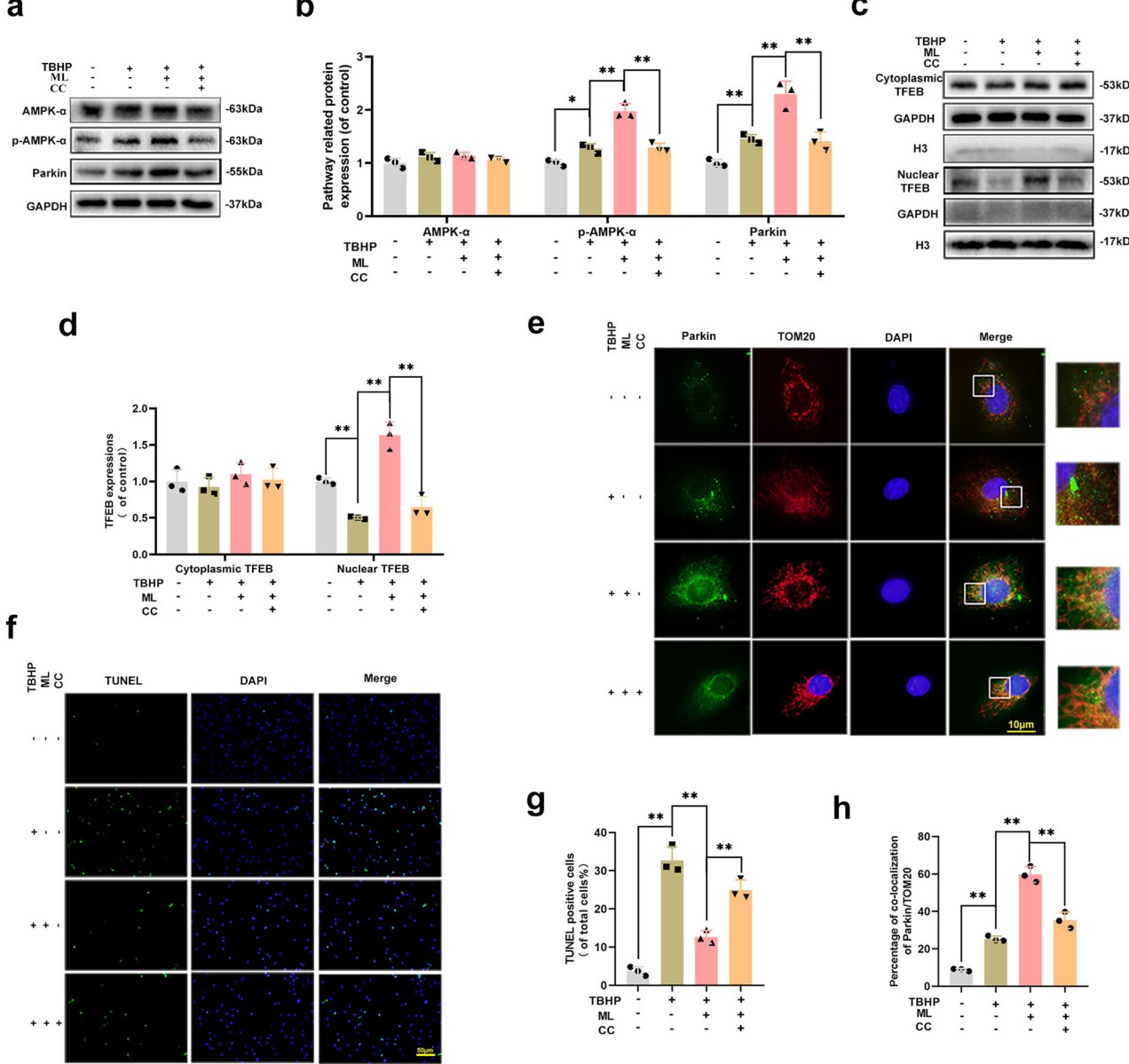

**Fig. 6 ML induced mitophagy via AMPK-TFEB signaling pathway in HUVECs. a, b** Expression and quantification of proteins AMPK-α, p-AMPK-α, and Parkin were measured by western blot. **c, d** Western blots were used to evaluate the level of cytoplasmic and nuclear TFEB in HUVECs. **e, h** Image of double-labeled staining of Parkin and Tom20 in HUVECs treated as above (Green: Tom20, red: Parkin, bar: 10 μm). **f, g** Cell apoptosis was evaluated using TUNEL assay in HUVECs (bar: 50 μm). All experiments have been performed at least 3 times (n ≥ 3). Data are presented as mean ± S.D. n = 3, **P < 0.01, *P < 0.05.

different trend from the protein mentioned above. In the ML group, the number of C-caspase3 positive cells declined compared with the control group. However, after CC treatment, the number of C-caspase3 positive cells increased. Furthermore, we measured the SOD, GSH, and MDA levels in tissues. SOD and GSH maintain cell homeostasis by promoting ROS clearance and are inversely correlated with oxidative stress levels[27]. MDA is a product of lipid peroxidation and is positively correlated with oxidative stress levels[28]. Higher levels of SOD and GSH but a lower level of MDA was observed in the ML group, compared with the control group. After co-administration with CC in the ML + CC group, the level of SOD and GSH were dramatically decreased. Conversely, the level of MDA increased (Fig. 8i–k). The molecular mechanisms of ML-induced mitophagy, anti-oxidative stress, and anti-apoptosis action are depicted in

schematic diagrams (Fig. 9). Taken together, these results imply that ML enhancing Parkin-dependent mitophagy through the AMPK-TFEB signaling pathway is a potential mechanism underlying the anti-apoptotic and anti-oxidative stress in promoting flap survival.

## Discussion
The random-pattern skin flap is universally used in reconstruction for wound covering and skin defects reparing[29,30]. However, flap necrosis is a main postoperative complication and limits clinical application[31]. Insufficient blood supply and subsequent I/R injury are considered the principal causes of flap necrosis[32,33]. When partial blood supply of the flap is restored, tissues may undergo I/R injury. As a result, an outbreak of ROS is generated. It has been

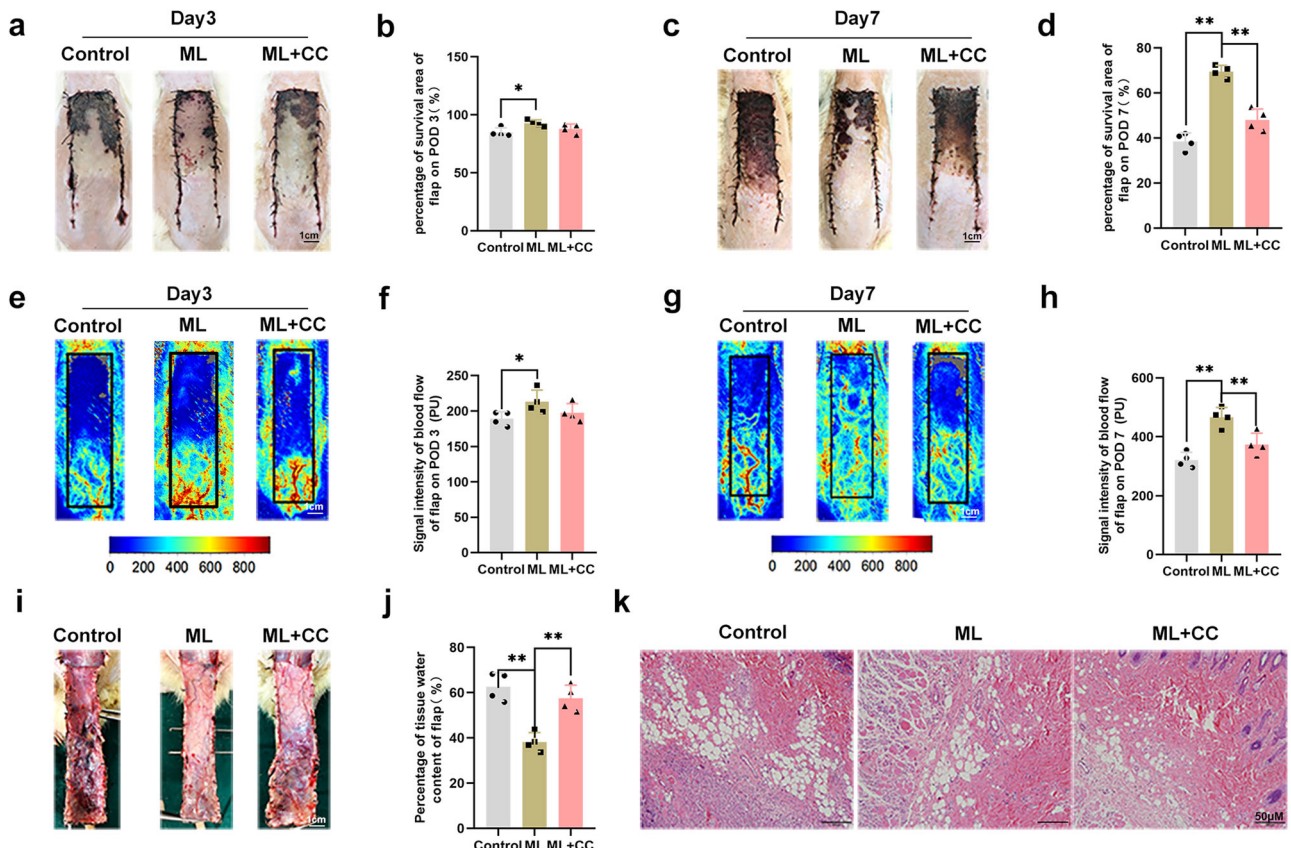

**Fig. 7 ML promoted random-pattern skin flap survival and was reversed by AMPK inhibitor. a**, **c** Digital photograph of the flap on POD 3 and POD 7 (bar: 1 cm). **b**, **d** Percentages of survival area on POD 3 and POD 7 were quantified and demonstrated by the histogram. **e**, **g** Laser Doppler was carried out to visualize the blood supply under the flap on POD 3 and POD 7 (bar: 1 cm). **f**, **h** Quantify the signal intensity of blood flow. The stronger the signal intensity, the greater the blood flow. **i** Image of tissue edema and Subcutaneous vascular network in control, ML, and ML + CC groups on POD7 (bar: 1 cm) ($n = 4$). **j** Percentage of tissue water in each group. **k** Representative histological images of skin flaps exhibiting blood vessels and inflammation (bar, 50 μm). Data are presented as mean ± S.D. $n = 4$ per group, **$P < 0.01$, *$P < 0.05$.

reported that different levels of ROS production exert different effects on cell survival. Specifically, an appropriate amount of ROS can activate autophagy and protect cells from oxidative stress, while excessive ROS production can lead to lipid peroxidation, apoptosis, and death. Thus, inhibiting ROS burst is a potential strategy to alleviate skin flap from I/R injury[34,35]. ML, a potent free radical scavenger, was reported to promote flap survival with its anti-inflammatory effects[36,37]. However, the specific mechanism of its anti-oxidative stress in enhancing skin flap survival remains elusive. Thus, we sought to explore the role of ML in inhibiting oxidative stress and its impact on the ROS level and the mechanism of action under the TBHP stimulation model.

The mitochondria is the main source of intracellular ROS production[38,39]. Oxidative stress injury can increase the permeability of the mitochondrial inner membrane and disrupt the electron transport chains[40–42]. The increased mitochondrial inner membrane permeability leads to the dissipation of MMP and the release of Cytochrome c into the cytoplasm, which eventually triggers caspase-mediated apoptosis[43,44]. Bi et al. reported that recovering MMP can reduce lymphocytic apoptosis[45]. Herein, the results showed that ML treatment prevented mitochondrial ROS production in vitro and the results of the JC-1 probe exhibited that MMP was largely recovered in the ML group. In addition, western blot measured levels of pro-apoptotic proteins Bax and C-caspase3 and the anti-apoptotic protein Bcl-2 and results demonstrated that ML treatment suppressed TBHP-induced apoptosis in HUVECs. Therefore, we established that ML could suppress mitochondrial dysfunction and apoptosis in HUVECs.

Mitophagy is mitochondrial quality control and can eliminate damaged mitochondria to avoid a destructive burst of ROS[46]. Lee et al. reported that ML suppresses mitochondrial dysfunction via the HSPA1L-mitophagy pathway[17]. However, there was no study on ML-induced mitophagy in HUVECs. To better explore whether mitophagy was involved, we administrated two types of mitophagy inhibitors, CsA and Mdivi-1. CsA could bind to cyclophilin protein D located in the mitochondrial matrix space, thereby specifically inhibiting mitochondrial permeability transformation and thus reducing the occurrence of mitophagy[47]. On the other hand, Mdivi-1 inhibits mitophagy by affecting the dynamic phosphorylation status of fission protein dynamin-related protein 1 (Drp1)[48]. Our results (Fig. 3 and Supplementary Fig. 3) confirmed that the inhibition of mitophagy induced the depolarization of MMP, provoked mitochondrial dysfunction, and triggered the accumulation of mitochondrial-derived ROS resulting in cytotoxic. Herein, we confirmed that ML activated mitophagy in HUVECs and inhibited ML-induced mitophagy lead to depolarization of MMP and mitochondrial dysfunction. In addition, when mitophagy was absent, oxidative stress and apoptosis level increased. Taken together, these results suggest that ML protects HUVECs from oxidative stress mainly by activating mitophagy.

Parkin is known to be highly involved in mitochondrial quality control[49]. ROS accumulation and mitochondrial disruption lead to the increase in PINK1 stability and the subsequent recruitment of Parkin[50,51]. Cao et al. described that ML could upregulate Parkin-mediated mitophagy, and reduce mitochondrial damage,

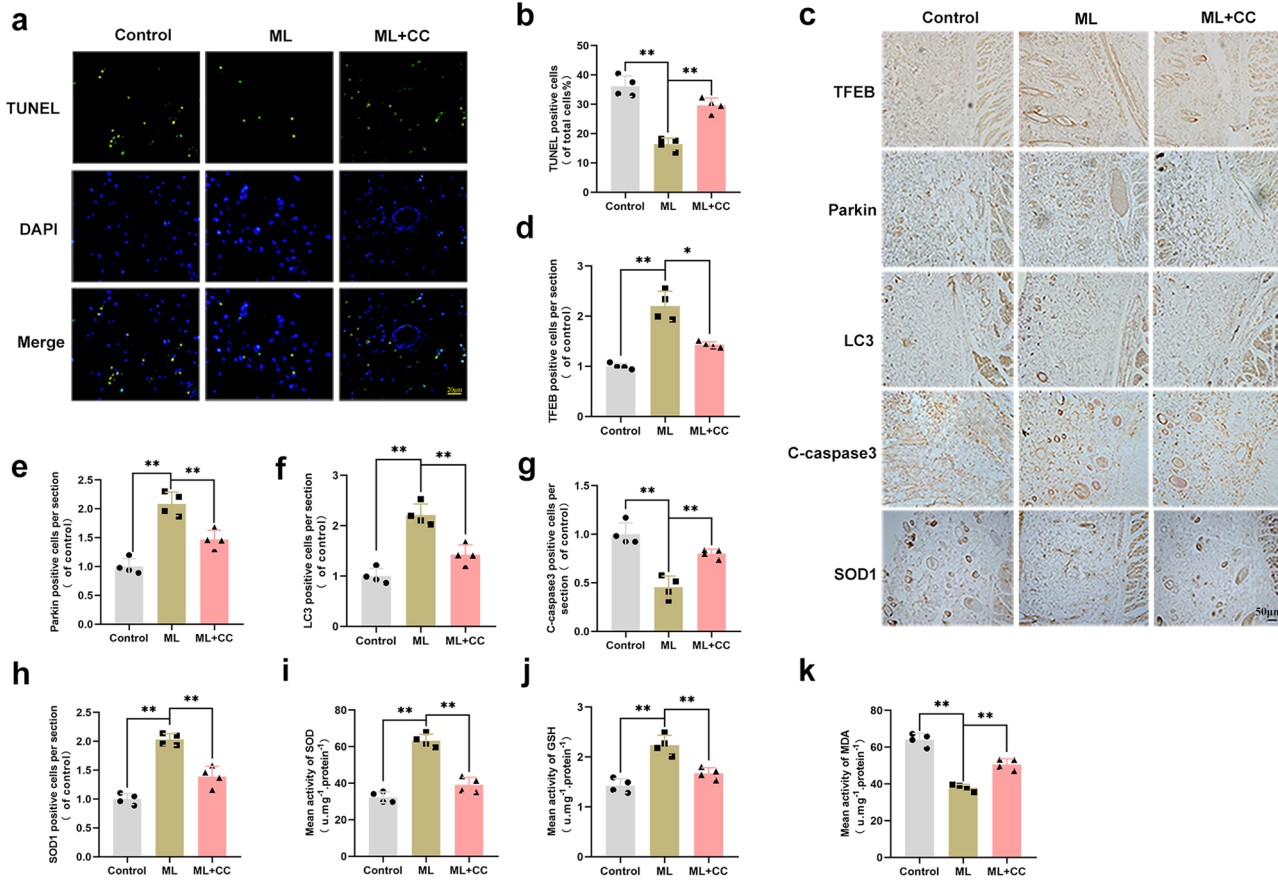

**Fig. 8 In vivo, ML reduced apoptosis and oxidative stress via inducing mitophagy. a**, **b**. TUNEL staining was performed to detect apoptosis in the flap in control, ML, and ML + CC groups (bar: 50 μm). **c–h** IHC results exhibited the expression of TFEB, Parkin, LC3, C-caspase3, and SOD1 positive cells in each group and the optical density values of these proteins were quantified (bar: 50 μm). **i** Total SOD activity was detected by xanthine oxidase method. **j** GSH level was assessed by modified 5,5'-dithiobis method. **k** MDA level by measured by thiobarbituric acid test. All experiments have been performed at least 4 times. Data are presented as mean ± S.D. $n = 4$ per group, $**P < 0.01$, $*P < 0.05$.

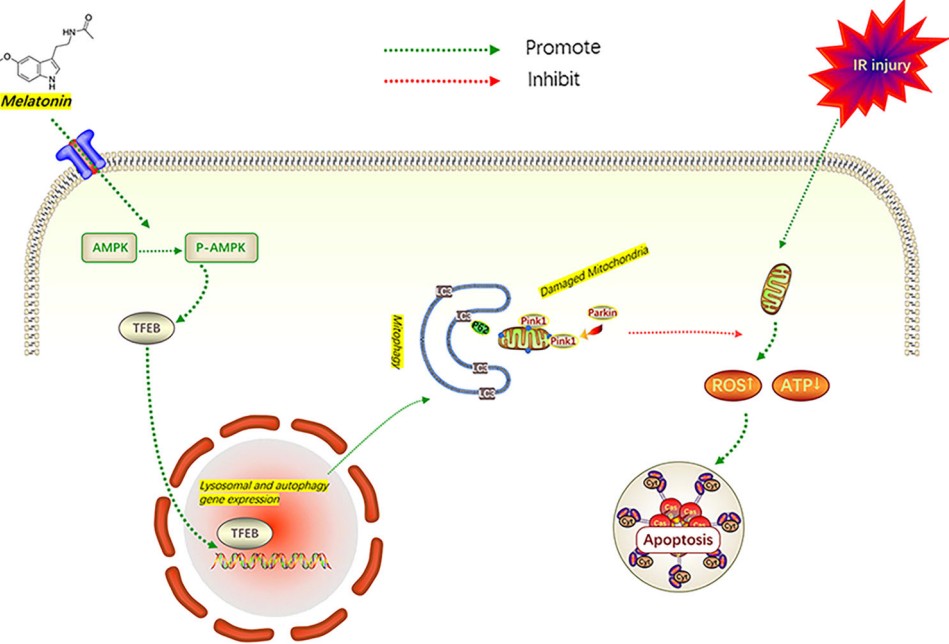

**Fig. 9 Potential mechanisms of ML on HUVECs and flap survival.** Schematic illustration showing ML alleviates apoptosis and oxidative stress injury via promoting Parkin-dependent mitophagy through the AMPK-TFEB signal pathway.

and generate ROS in the subarachnoid hemorrhage model[52]. These suggest that there is a cross-reaction between ROS and Parkin. Hence, we hypothesized that the protective effects of ML in HUVECs might also through Parkin-mediated mitophagy. Our sequencing and western blot results found that Parkin was upregulated. However, when Parkin was knocked down by si-RNA, the fluorescence intensity of Mitotracker was decreased. It is interesting that mitophagy should normally prevent the degradation of mitochondria and thus increase the number of mitochondria. This may be due to the intensity of Mitotracker fluorescence is related to the level of mitochondrial membrane potential[53,54]. Thus, we applied the transmission electron microscopy which not depend on the functional status of the mitochondria and can directly exhibit the number and morphology of mitochondria. The result demonstrated that in si-Parkin depletion caused mitochondrial vacuolation, shrinkage, rupture, and fewer mitophagosomes in the si-Parkin group, which indicated that Parkin inhibition markedly declined the mitophagy level and the protective effect of ML was largely abolished. Altogether, our data demonstrated that Parkin-dependent mitophagy might be the principal mechanism of the therapeutic effects of ML in HUVECs.

In order to elucidate the role of ML in mitophagy, it is imperative to further explore the mechanism of Parkin upregulation. Preceding studies have proved that AMPK, an energy-sensing protein kinase, is associated with Parkin regulation[55,56]. Activation of the AMPK signaling pathway can suppress mitochondrial dysfunction[57]. TFEB, substantially promoted autophagy and the transcriptional regulation of lysosome-related genes[58,59]. Moreover, AMPK is considered necessary for the phosphorylation of TFEB[60–62]. Therefore, we hypothesized that the AMPK-TFEB pathway is involved in ML-induced mitophagy. According to our sequencing and western blot results, we found that TFEB was indeed involved in the regulation of mitophagy by ML (Figs. 4 and 5). In addition, western blot results showed that ML treatment enhanced the phosphorylation of AMPK-α but inhibited AMPK-α partly decreasing the expression level of nuclear TFEB and Parkin. Besides, inhibited TFEB by si-RNA also abolished the anti-apoptotic and anti-oxidative stress effects of ML. Meanwhile, we also revealed that the number of TFEB and Parkin-positive cells were surged after ML was administrated in vivo. However, when AMPK was inhibited the anti-oxidative stress and anti-apoptotic ability were suppressed. Taken together, these data implied that the AMPK-TFEB signaling pathway might be responsible for the Parkin-mediated mitophagy triggered by ML.

Notwithstanding the novel findings in the present work, certain limitations are worth acknowledging. First is the side effect of Compound C. When Compound C inhibits AMPK, it also inhibits other kinases such as ERK8, MNK1, and PHK[63]. Therefore, whether these kinases affect AMPK expression is still unclear. An AMPK knockdown rats' model might be better. Second, the regulation of Parkin at the gene level can better validate the role of ML in skin flap. In this case, the Parkin +/+ and Parkin −/− mice should be used in further studies. Moreover, angiogenesis also plays an essential role in the survival of flaps, and our study focused on exploring the anti-oxidative stress mechanism of ML; only a few experiments regarding angiogenesis were performed (Fig. 7). Thus, the effect of ML on skin flap angiogenesis should be noted in further studies.

In summary, we firstly demonstrated the relationship between the AMPK-TFEB signaling pathway and Parkin-mediated mitophagy in random-pattern skin flaps. Moreover, our results exhibited that the anti-oxidative stress and anti-apoptotic effects of ML were greatly associated with the activation of Parkin-mediated mitophagy. These findings highlight the role of mitophagy in flap survival and provide strong evidence that ML promotes flap survival.

## Methods

**Animals and ethics statement**. In total 60 male Sprague-Dawley rats (8-week-old, 200–250 g) were received humanitarian care, and the experimental procedures were in accordance with the guidelines of the Animal Care and Use Committee of Wenzhou Medical University (ethics code: wydw 2021-0256).

**Reagents and antibodies**. Melatonin, Bafilomycin A1, mitochondrial division inhibitor 1 (Mdivi-1), Urolithin A (UA), and Compound C (CC) were obtained from Med Chem Express (Monmouth Junction, NJ, USA). Tert-Butyl hydroperoxide solution (TBHP) and Cyclosporin A (CsA) were acquired from Sigma-Aldrich (St Louis, MO, USA). Antibodies against Heme Oxygenase 1 (HO-1), LC3, and P62 were purchased from Abcam (Cambridge, MA, USA). Antibodies against Cleaved caspase-3 (C-caspase3), PINk1, Parkin (PRKN), AMPK-α (t-AMPK-α), and phospho-AMPK-α (p-AMPK-α) were purchased from Cell Signaling (Beverly, MA, USA). Antibodies against Superoxide Dismutase 1 (SOD1), Bax, Bcl-2, GAPDH, Transcription factor EB (TFEB), and Histone-H3 (H3) were obtained from the Proteintech Group (Chicago, IL, USA). Malondialdehyde (MDA), glutathione (GSH), and superoxide dismutase (SOD) kits were obtained from Nanjing Jiancheng Biology Jiancheng Technology Institution.

**Cell culture**. As previous studies reported, to simulate apoptosis and oxidative stress injury in vitro experiments of skin flaps, Human umbilical vein endothelial cells (HUVECs) were selected[64,65]. HUVECs were obtained from ATCC (Manassas, VA) and cultured with Dulbecco's modified eagle medium (Gibco, Carlsbad, USA), 10% fetal bovine serum (FBS, Gibco, Carlsbad, USA), and 1% penicillin and streptomycin (Gibco, Carlsbad, USA) in a humidified incubator with 5% $CO_2$ at 37 °C. After reaching 70–80% confluence, HUVECs were treated with the corresponding treatment and used for subsequent experiments.

**Experimental design**. In vitro, tert-butyl hydroperoxide (TBHP) was applied to induce oxidative stress in HUVECs, given the stable and sustained-release characteristics of TBHP compared to $H_2O_2$. To explore the role of ML activating mitophagy, UA (40 μM), a powerful mitophagy inducer, was used as a positive control in this experiment. Next, to verify the role of ML-induced mitophagy in HUVECs, CsA (1 μM) and Mdivi-1 (1 μM) as mitophagy inhibitors were administrated 1 h before ML treatment. Afterward, a siRNA was transfected to investigate the role of parkin in ML-induced mitophagy. According to the cell sequencing result, CC (40 μM, an AMPK inhibitor) and si-TFEB were used to study the AMPK-TFEB pathway.

In vivo, 60 rats were assigned to four groups; the sham group (n = 6) and control group (n = 18) were treated (i.p.) with vehicle only. In contrast, the ML group (n = 18) was treated with 40 mg/kg ML 1 h before the operation and continued for 6 days[37]. The ML + CC group (n = 18) was treated with CC 1.5 mg/kg 30 min before ML administration every time[66]. At POD 3, all rats were anesthetized and scanned by Laser Doppler. At POD 7, the rats were first scanned by Laser Doppler and then euthanized with an anesthetic overdose, and tissue samples from area II were collected for western blot, histological, tissue edema analyses, and analyses of oxidative stress levels.

**TUNEL staining**. Apoptotic DNA fragmentations of HUVECs and flap sample sections were detected by TUNEL staining. According to the manufacturer's instructions, cells were fixed with 4% paraformaldehyde for 30 min after treatment, then incubated with 3% $H_2O_2$ and 0.3% Triton X-100 for 10 min. Next, HUVECs were stained with the TUNEL Kit (Beyotime Biotechnology, China) for 1 h at 37 °C. After washing with PBS three times, the nuclei were stained with DAPI. Finally, TUNEL positive cells were observed under an Olympus fluorescence microscope (Olympus Inc., Tokyo, Japan).

**Mitochondrial membrane potential (MMP)**. The JC-1 fluorescent probe (Beyotime Biotechnology, China) was used to monitor the alterations in MMP. Briefly, HUVECs were incubated with JC-1 dye for 30 min at 37 °C after treatment and washed three times with the JC-1 buffer solution. According to the manufacturer's instructions and the previous study[67,68], a fluorescence microscope was used to visualize red and green fluorescence, and the intensity was analyzed by Image-Pro Plus 6.0 (Media Cybernetics, MD, USA).

**MitoTracker and MitoSOX Red statin**. The level of mitochondrial ROS was detected by MitoSox Red dye (Invitrogen, M36008) and mitochondrial distribution was detected by MitoTracker Red (Invitrogen, M24426). After the indicated treatments, HUVECs were incubated with MitoSox dye (20 μM) or MitoTracker dye (50 μM) for 30 min at 37 °C. After three times washing with PBS, Hoechst was used to stain the nuclei at 37 °C. A fluorescence microscope was utilized to capture images.

**ATP assay**. ATP levels reflect mitochondrial energy production and alterations in ATP levels affect cellular function. The ATP assay kit (beyotime, S0026) was used to measure the mitochondrial energy production in HUVECs. In short, 100 microliter ATP test solution was added to HUVECs lysis buffer at room

temperature for 5 min. Luminescence was measured by the Multiskan™ Sky Microplate Spectrophotometer (Thermo, USA).

**Cell viability assay**. The effect of TBHP, ML, and CsA on HUVECs viability was detected by CCK8. Firstly, approximately 5000 cells were placed into a 96-well plate for 24 h. Then different concentrations of TBHP (0, 12.5, 25, 50, 100 and 200 μM), ML (0, 1, 10, 100, and 1000 μM) and CsA (0, 0.1, 1, 10, and 100 μM) were administrated for 24 h. After three times washing with PBS, 10 μL CCK-8 solution and 90 μL DMEM were mixed into each well and incubated for 1 h. The absorbance was measured by Multiskan™ Sky Microplate Spectrophotometer (Thermo Scientific, USA) at 450 nm.

**Western blot**. Total cellular proteins from HUVECs were extracted by RIPA lysis buffer with 1 mM PMSF. Nuclear and cytoplasmic proteins were separated by the nuclear and cytoplasmic protein extraction kit (Beyotime Biotechnology, China) following the manufacturer's protocol. Next, a BCA kit (Beyotime Biotechnology, China) was used to measure protein concentrations. Proteins were subjected to electrophoresis and transferred to PVDF membranes. (Millipore, Bedford, MA, USA). After blocking by NcmBlot blocking buffer (New cells & Molecular Biotech, China), the membranes were incubated with the corresponding antibodies overnight at 4 °C. Then these membranes were incubated with the secondary antibodies for 2 h. The Image Lab 3.0 software (Bio-Rad, California, USA) was employed to quantitate and analyze blots' grayscale values.

**Immunofluorescence**. HUVECs were seeded on glass-bottom dishes and treated with the corresponding indicated agents. After fixation, cells were permeabilized with Triton X-100 for 10 min at room temperature. Then, the cells were blocked with goat serum albumin for 1 h and incubated with the primary antibodies Parkin (1:200), LC3 (1:200), and Tom20 (1:200) overnight at 4 °C. After washing three times, the samples were incubated with FITC-conjugated secondary antibody for 1 h at 37 °C. Nuclei were stained with DAPI solution. A fluorescence microscope was used to capture images and the integrated optical density was analyzed by the Image-Pro Plus software.

**Histomorphology and Immunohistochemistry (IHC) analyses**. Samples from Area II were fixed, paraffin-embedded, and sliced into 5 μm thick slides. Then, the hematoxylin and eosin (H&E) Kit (Solarbio Science & Technology) was used to stain the slides.

For IHC, slides were first deparaffinized and rehydrated. After blocking thec endogenous peroxidase, these slides were incubated with primary antibodies against LC3 (1:100), Parkin (1:200), TFEB (1:100), SOD1 (1:200), HO-1 (1:200) overnight at 4 °C. After washing three times, these slides were incubated with a secondary antibody for 1 h at 37 °C. Finally, images were captured under an optical microscope and analyzed with the Image Pro-plus software.

**Flap model**. Rats were anesthetized by injecting pentobarbital sodium solution (40 mg/kg). The hair on the dorsal of rats was removed with a shaver and depilatory cream. Then, a $3 \times 9$ cm$^2$ area was marked on the rat's back, and the flap was dissected and elevated from the rat dorsum. Afterward, the bilateral arteries at the bottom of the flap were cut, and latter was sutured in situ with 4-0 silk. The skin flap was artificially divided into three equal parts, from the bottom to the head: Area I, Area II, and Area III. In the sham group, only the skin was cut and sutured, but bilateral arteries were not cut off. The difference in the survival rate of rat skin flap was mainly shown in Area II. Therefore, the tissue of Area II was selected for the following experiments.

**Flap survival area and tissue edema assessment**. On POD 3 and POD 7, the flap on the rat's back was photographed and the survival area was analyzed by the ImageJ software (National Institutes of Health, Bethesda, MD).

The water content in the skin flap can indirectly reflect the amount of edema. On POD 7, rats were euthanized and the whole flap was harvested. The flaps were subsequently placed in can autoclave and weighed until the weight was stable.

**Laser Doppler Blood Flow Measurement**. Living rats' flap blood flow was detected by Laser Doppler. On POD 3 and POD 7, rats were scanned by a Laser Doppler instrument (Moor Instruments, Axminster, UK) under anesthesia. Blood flow was quantified using the perfusion unit and calculated using the Moor LDI Review software (ver.6.1; Moor Instruments).

**RNA-sequencing and bioinformatics analysis**. HUVECs were grown to 70–80% confluence and divided into three groups: Control, TBHP, and TBHP + ML groups. The cells received the corresponding treatment, and then the TRIzol reagent (Invitrogen, 15596018) was used to extract the total RNA. RNA-sequencing was performed as literature mentioned[69]. Bioinformatics analysis was performed using BMKCloud (www.biocloud.net).

**Transmission electron microscopy (TEM)**. HUVECs were fixed with an electron microscope fixing solution (Servicebio, China) for 2 h at room temperature. After the cells were dehydrated and embedded, semithin sections were cut. Lastly, images were captured under a transmission electron microscope (Hitachi, Tokyo, Japan).

**Measurement of oxidative stress levels**. SOD, GSH and MDA kits were utilized to measure the oxidative stress levels of the flaps. According to the manufacturer's instructions and literature described methods[70], samples from Area II in each group were homogenized, centrifuged, with the supernatant separated supernatant for analyses.

**Mitophagy detection assay**. The mitophagy detection kit (Dojindo Co, Japan) was used to detect mitophagy in HUVECs. In brief, HUVECs were seeded in a six-well plate covered with slides and incubated with 100 nM Mtphagy Dye solution for 30 min. After washing three times, HUVECs were incubated with ML and UA for 6 h. Next, the cells were washed three times and incubated with 1 μM LysocDye solution for 30 min. Eventually, a fluorescence microscope was used to capture images, which were subsequently analyzed by the Image-Pro Plus software.

**Statistics and reproducibility**. All experiments were repeated at least three times. Statistical analyses were performed using the SPSS statistical software package (version 22.0; Chicago, IL, United States). All statistical analyses were performed by one-way analysis of variance (ANOVA) with Tukey's multiple comparisons test to assess differences among the groups. $P < 0.05$ was considered statistically significant.

**Ethics statement**. Experimental procedures involving animals complied with the Guide for Care and Use of Laboratory Animals of the China National Institutes of Health, with acceptance of the Animal Care and Use Committee of Wenzhou Medical University (wydw 2017-0022).

**Reporting summary**. Further information on research design is available in the Nature Research Reporting Summary linked to this article.

## Data availability

The raw data supporting the conclusions of this article will be made available by the authors, without undue reservation, to any qualified researcher. The source data for original uncropped blot/gel images are provided in Supplementary Information. Uncropped Western blot images are provided in Supplementary Fig. 5. All relevant data including the numerical and statistical source data that underlie the graphs in figures are provided in Supplementary Data 1. We have uploaded the raw and processed data of RNA sequencing data in Gene Expression Omnibus. The GEO number is GSE203609.

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

## Acknowledgements

This work was supported by grants from China National Natural Science Foundation (81873942).

## Author contributions

Z.C., H.W., J.Y., B.L., S.C., and C.Z., performed experiments and analyzed data. H.L., D.L., and W.G. proposed and supervised the project and wrote the manuscript. J.D., N.B., and J.L. help improve the English presentation. All authors confirmed and edited the manuscript.

## Competing interests

The authors declare no competing interests.
