## [Peer Review File · Communications Biology]

Reviewers' comments:

Reviewer #1

In this study, Chen et al. study the role of melatonin as a mitophagy inducer improving random-pattern skin flaps survival through parkin-dependent reduction of oxidative stress. This is an interesting study which suffers from several limitations.

First, the rationale of the study is not clear. This should be more extensively detailed in the introduction section. Namely, why do the authors focus on mitophagy? It is not stated in the introduction that the induction or the inhibition of mitophagy is a key determinant of skin flap outcome. In line with this, the author performed *in vivo* skin flap experiments but do not show that there is a differential modulation of mitophagy between injured and healthy skin areas upon skin flap. This should be shown as it is one of the basis of the study. Another important issue is that the authors claim that the mechanism of action of melatonin (ML) is Parkin-dependent but it was not proved *in vivo* by performing skin flap experiments with skin from Parkin +/+ and Parkin -/- mice. The author used melatonin (ML) to induce mitophagy. Again, the rationale for this choice is not clear as the scientific literature with regards to the therapeutic interest of ML in the context of skin flap is not vast (Gurlek et al. *J Pineal Res.* 2004 Jan;36(1):58-63 only?). The assessment of a new therapeutic strategy targeting mitophagy is of interest. Again, the author should better justify the rationale for this approach. Additionally, they should bring more clues about the ability of ML to promote mitophagy. Since ML is not a specific mitophagy inducer, this proof of concept should be strengthened by the use of additional mitophagy inducers *in vitro* and *in vivo* such as urolithin A which is one of the best characterized mitophagy inducers so far (e.g. PMID: 27400265). Similarly, the authors should use additional pharmacological inhibitors of mitophagy since cyclosporine A is not the best characterized compound in the field. Mdivi-1 should also be tested to bring more evidence regarding the mechanism of action of ML. Additionally, genetically-deficient mice should be used to further reinforce the proposed mechanism especially Parkin and AMPK-deficient mice.

Other points to be addressed:

Major points

The authors should justify the choice of HUVEC cells to investigate the mechanism of action of ML *in vitro*

The authors should systematically assess the impact of the compounds alone and in combination. In many experiments, the impact of ML and CsA exposure is only assessed in combination.

The authors claim that they measure mitochondrial dysfunctions but, a more comprehensive approach should be employed (not only EM): assessment of metabolism + measurement of mtDNA damage + measurement of oxidized cardiolipin for example. Additionally, p9, the sentence "TEM images showed a greater incidence of mitochondrial damage" is an overstatement.

The authors should use a quantitative approach (such as flow cytometry) to assess mitochondrial density, MMP and mROS in addition to microscopy. With this regards, the methodology of quantification of IF images should be better explained. More generally, it is not clear how the stats were made for microscopy approaches.

The authors should use dedicated technology such as mitoKeima to quantify mitophagy

Minor

There are numerous spelling mistakes in the manuscript. Additionally, wording is sometimes surprising or vague e.g. p4 "overdose of anesthetic", p9: "Parkin is essential for mitophagy on most occasions", p10: "the four groups were not significant".

Use colors rather than grayscale in the figures. The bar graphs are not always easy to distinguish. The order of the panels should be re-arranged to follow the flow of the text.

Reviewer #2 (Remarks to the Author):

In their article entitled « Activating Parkin-dependent mitophagy alleviates oxidative stress and apoptosis in human umbilical vein endothelial cells and promotes random-pattern skin flaps survival », Chen et al. use an *in vitro* model system to dissect the molecular mechanisms responsible for the improvement of random-pattern skin flap survival upon melatonin treatment. To model the ischemia/reperfusion injury that is thought to occur upon random-pattern skin flap surgery, they submit HUVEC to oxidative stress *in vitro* (TBHP treatment). They claim that, in these conditions, melatonin (ML) treatment alleviates cell death by triggering mitophagy. They observe an involvement of Parkin and of the AMPK/TFEB pathway. Finally, they conduct *in vivo*

experiments to confirm their in vitro findings.

Although the novelty of the conclusions is not major (the effect of ML on skin flaps as well as the effect on ML on mitophagy have been described previously in reference #13 and 14 as the authors themselves acknowledge), the study is well conducted and brings new light on the role of melatonin in the context of random-pattern skin flaps survival. There are, however, several points that need to be addressed before publication:

Major points

1) The authors claim that autophagy is increased in TBHP+ML conditions as they observed varying levels of the autophagy markers LC3II and p62. However, an increase in LC3II could be due to an increase in autophagy induction or a decrease of autophagic degradation. The authors should use a lysosomal inhibitor such as Bafilomycin A1 to block autophagic flux and to study whether their LC3II increase really reflects an increase in autophagy activation. The same is true for the decreased levels of p62 upon ML treatment, which could reflect an increase in autophagy or a decrease in oxidative stress, which is known to regulate p62 at the transcriptional level (pmid:20452972). The use of Bafilomycin A1 treatment on the in vitro HUVEC system is crucial to conclude here.

2) As one of the main conclusions of the study is that increasing mitophagy promotes random-pattern skin flaps survival, the authors should try to increase mitophagy in their system other than by ML treatment in order to rule out that what they observe is not due to a more general antioxidant effect of ML. Mitophagy activators have recently been described (PMCID: PMC7665171 for example).

Minor points

1) Although it does not impact the general understanding of the results presented, significant rewrites should be made to improve the fluidity of the text. Some sentences are not very precise and need to be put in context. For instance: "Moreover, ATP production analysis suggested that ML enhanced mitochondria-derived ATP and inhibited mitophagy and blocked ML's effect on mitochondria energy production (Fig. 3C)" (line 244).

2) line 234: Fig.2F does not exist. Co-localization studies should be quantified as they don't appear striking on the images alone (this is also true for FigS2).

3) The RNA-seq results are not well presented: where are the genes in Fig.4B on the Fig.4A plot? Presenting the sequencing data in a table should also be considered.

4) Please explain the effect of si-Control on the phenotypes studied (Figure 4).

5) Please show the efficiency of Nuclear/cytoplasmic fractionation (Figure 5).

6) Several errors could be spotted in the text:

- line 227: Fig.2E, F should be replaced by Fig.2C, D
- line 406: "Founding" should be replaced by "Funding"
- line 572: "treated as above" should be elaborated.

7) Several errors could be spotted in the figures:

- FigS1B: μm should be replaced by μM
- Fig3A: western blots should be labeled PINK1 and not PINK
- Fig3G: TBHP+ML+CC should be replaced with TBHP+ML+CsA
- Fig3F: red and green panels were inverted (left panels)
- Fig3: panels should be re-ordered as to show 3C as the last panel of the figure
- text in Fig4A and Fig4B is way too small (see minor comment #3)
- Fig4I: replace "enlarged" with "higher magnification"
- Fig5C: does not seem to be the same format as the other panels. "TFEB" should be replaced with "TBHP"
- Fig7A: please show insets as labeling is difficult to see

- Fig8: explain the red arrow (should it be inhibitory?)

Reviewer #3 (Remarks to the Author):

The study of Zhengtai Chen and co-workers aimed at investigating the mechanism of mitophagy induced by Melatonin and its effect on the survival of skin flaps. From their results, the authors concluded that Melatonin induced mitophagy in human umbilical vein endothelial cells through a mechanism that involves Parkin 1. The induction of mitophagy alleviates oxidative stress and apoptosis. In in vivo study, the authors found that melatonin promotes mitophagy to enable flap survival. Overall, the manuscript is well-written and the topic is original. However, the authors have overinterpreted some data and additional studies are needed to reinforce the data of the manuscript.

Below are my specific concerns on the manuscript:

I) My major concern is related to the induction of mitophagy by Melatonin.

Figure 2A: a significant accumulation of LC3-II was observed in cells treated with TBHP+Melatonin versus TBHP alone but it is unclear if this accumulation is related to the induction or inhibition of (mito)autophagy. If mitophagy is functional, the expression of LC3-II should decrease as LC3-II is degraded along this process. To validate the induction of mitophagy, experiment presented in the figure 2A should be repeated in the presence and absence of Bafilomycin A1 (an inhibitor of lysosomal function). Otherwise, I suggest to use the Mito-Keima tandem plasmid which is a pH sensitive fluorescent protein located in mitochondria that allows the detection of mitochondria movement from cytosol to lysosomes.

Figure 2C: To validate mitophagy, the number of mitochondria should be counted after staining cells with MitoTracker dye in the presence and absence of Bafilomycin A.

Figure 2E: A better resolution of images is needed to appreciate the co-localization of LC3 and TOM20 in cells treated with TBHP+ML. Please note that LC3 and TOM20 staining were both increased under TBHP + ML. Can the authors provide an explanation for this observation?

II) Figure 3A: the authors used CsA as a mitophagy inhibitor and found that inhibition of autophagy led to a decrease of both LC3-II and p62 expression levels. This is puzzling as normally these proteins are accumulated during inhibition of mitophagy. Can the authors provide an explanation for this observation?

Figure 3F: quantification is needed for MPP.

Figure 4: To support the author's conclusion that Parkin 1 is involved in mitophagy, the authors should determine the effect of siParkin 1 on the mitochondria number (through staining with MitoTracker) and the expression levels of LC3-II and p62, as well. Moreover, it would be nice to have two siRNAs against Parkin 1 to confirm the results shown in Figure 4.

Figure 5: Figure 5A: please add the expression level of GAPDH in nuclear fraction and vis versa of H3 in cytoplasmic fraction (to show the efficiency of nuclear/cytoplasmic extraction). Moreover, it would be nice to confirm the subcellular localization of TFEB by immunofluorescence study in the experiments presented in figures 5A and 5B.

III) The authors should avoid over-interpretation of their "TFEB" results. The authors stated that "TFEB was indeed involved in the regulation of mitophagy by ML" (Line 383) but there is no evidence in the paper for such regulation. To support this statement, I suggest to examine the effect of TFEB si/shRNA on mitophagy-induced by Melatonin. Otherwise, the authors should rewrite the discussion part and change the figure 8 in accordance with their results.

Minor concerns:

- A better resolution of images is needed for Fig 2A (c-caspase 3) and Fig 3A (p62)

- Figure 5 C: the authors mistakenly wrote TFEB instead of TBHP

In my opinion, the data are not strong enough to warrant the conclusions reached by the authors. This study requires further evidence to support the induction of mitophagy by melatonin and its regulation.

Reviewer #1

In this study, Chen et al. study the role of melatonin as a mitophagy inducer improving random-pattern skin flaps survival through parkin-dependent reduction of oxidative stress. This is an interesting study which suffers from several limitations.

Response: We appreciate very much for your careful review and giving us a chance to revise the manuscript. The following are our answers to your questions. We hope it will satisfy you.

1. First, the rationale of the study is not clear. This should be more extensively detailed in the introduction section. Namely, why do the authors focus on mitophagy? It is not stated in the introduction that the induction or the inhibition of mitophagy is a key determinant of skin flap outcome.

Response: Thanks for your careful review. We made the explanation on page 2, lines 44-47 and 54-59. Please check it.

After neovascularization, restoration and reperfusion of the blood supply trigger ischemia-reperfusion injury of flaps¹. During the process of I/R accumulation of reactive oxygen species (ROS) can respond non-specifically and promptly with cellular biomolecules, including DNA and proteins which eventually trigger DNA variations, protein oxidation, and lipid peroxidation². The Mitochondria within cells are a major source of endogenous ROS³. Besides, mitophagy is a selective macroautophagy that can remove damaged mitochondria and avoid a destructive burst of ROS⁴. Moreover, it was reported that mitophagy exerted protective functions in various pathological processes of several diseases, including spinal cord ischemia-reperfusion injury⁵, acute kidney injury⁶, and oxidative stress-induced intestinal barrier injury⁷. Thus, activating mitophagy might be a powerful approach to reduce oxidative stress injury during flap translation.

- 2.

In line with this, the author performed in vivo skin flap experiments but do not show that there is a differential modulation of mitophagy between injured and healthy skin areas upon

skin flap. This should be shown as it is one of the basis of the study

Response: Thank you very much for your suggestion. This data indeed is one of the bases of our study. Thus, we remeasured the level of mitophagy between the injured and healthy skin areas in Fig. 1A-C. Please kindly check it.

3. Another important issue is that the authors claim that the mechanism of action of melatonin (ML) is Parkin-dependent but it was not proved in vivo by performing skin flap experiments with skin from Parkin +/+ and Parkin -/- mice.

Response: Many thanks. Parkin +/+ and Parkin -/- mice are indeed a great way to verify the regulation of Parkin by ML and it is helpful to our study. However, it takes a long time to establish Parkin +/+ and Parkin -/- mice and the cost is too high for our group to afford. And our study mainly focused the mechanism of ML upregulated Parkin in vitro, so only functional verification was carried out in vivo. Therefore, this is a limitation of our study. We will further study the mechanism of Parkin-related mitophagy in the in vivo model of skin flaps. We have discussed it and marked in red on page 15, lines 453-455.

4. The author used melatonin (ML) to induce mitophagy. Again, the rationale for this choice is not clear as the scientific literature with regards of the therapeutic interest of ML in the context of skin flat is not vast (Gurlek et al. J Pineal Res. 2004 Jan;36(1):58-63 only?) The assessment of a new therapeutic strategy targeting mitophagy is of interest. Again, the author should better justify the rationale for this approach.

Response: We appreciate it very much for your kind suggestion. We have explained the rationale for the reason that we choose to study ML-induced mitophagy in skin flaps on page 3, lines 64-69 and marked in red.

First, melatonin is the natural secretion of the human body and has been found in almost all organisms⁸. Thus, the study of melatonin is promising. Second, the anti-inflammatory and

antioxidant stress effects of melatonin can effectively combat complications in the process of skin flap transplantation⁹⁻¹¹. Third, in recent years, increasingly studies have been conducted on ML's role in mitophagy, and some studies have found that ML up-regulated mitophagy can resist I/R damage. However, there is no literature on the regulation of mitochondrial autophagy by ML in skin flaps. Therefore, our group hypothesized that ML-induced mitophagy may also play an important role in flap survival. In summary, these are the reasons why we chose ML for this study.

5. Additionally, they should bring more clues about the ability of ML to promote mitophagy. Since ML is not a specific mitophagy inducer, this proof of concept should be strengthened by the use of additional mitophagy inducers in vitro and in vivo such as urolithin A which is one the best characterized mitophagy inducer so far.

Response: Thank you for your valuable advice. According to your suggestion, we used urolithin A as a positive control. Figure 2 has been reversed and the results part has been rewritten on page 8-9, lines 241-273. Please help us check.

6. Similarly, the authors should use additional pharmacological inhibitors of mitophagy since cyclosporine A is not the best characterized compound in the field. Mdivi-1 should also be tested to bring more evidence regarding the mechanism of action of ML.

Response: Thank you for your valuable advice. Mdivi-1 is a special mitophagy inhibitor and inhibits mitophagy by different mechanisms compared with cyclosporine A. Thus, we applied Mdivi-1 to inhibit mitophagy and found that Mdivi-1 successfully inhibited the ML-induced mitophagy and largely abolished the anti-oxidative stress and anti-apoptotic effects of ML. In addition, we also discussed the differences between Cyclosporine A and Mdivi-1 in the discussion section. Please check it in Fig. S3A-D; page 10, lines 285-288 and discussion section on page 14, lines 413-418.

7. Additionally, genetically-deficient mice should be used to further reinforce the proposed mechanism especially Parkin and AMPK-deficient mice.

Response: Thank you again for your valuable advice. The level of validation in the Genetically-deficient mice is indeed higher and more convincing. We have answered this question above in question 3. Our answers are displayed again below for your convenience.

To establish Parkin +/+, Parkin -/- and AMPK-/- mice are a great way to verify the regulation of Parkin by ML and it is helpful to our study. However, it takes a long time to establish Parkin +/+ and Parkin -/- mice and the cost is too high for our group to afford. Therefore, this is a limitation of our study. We will further study the mechanism of Parkin-related mitophagy in the in vivo model of skin flaps. We have discussed it and marked in red on page 15, lines 452-454.

8. The authors should justify the choice of HUVEC cells to investigate the mechanism of action of ML in vitro

Response: Thank you very much. Previous studies have shown that increasing angiogenesis and reducing ischemia-reperfusion (I/R) injury can reduce the necrotic area of the skin flap^{12,13}. HUVEC cells are often used to study vascular diseases. Moreover, as previous study reported that HUVECs were selected as cells to simulate apoptosis and oxidative stress injury in vitro experiments of skin flaps¹⁴⁻¹⁷. Altogether, we choose HUVEC cells to investigate the mechanism of action of ML in vitro. According to your suggestion, we also added the reason and reference in the manuscript on page 3, lines 88-89.

9. The authors should systematically assess the impact of the compounds alone and in combination. In many experiments, the impact of ML and CsA exposure is only assess in combination.

Response: Thank you for your careful check about our manuscript. First, Our CCK8 results showed that CsA did not affect cell survival at the concentration of 1 μ M (Fig. S1C). At the same time, we reviewed relevant literature, and no change in experimental results was reported on the use of CsA to inhibit mitophagy^{18,19}. Moreover, according to your suggestion, we also used a specific mitophagy inhibitor Mdivi-1, and the mitophagy inhibition efficiency of CsA was close to it. Therefore, please allow us not to conduct a separate study on CsA.

10. The authors claim that they measure mitochondrial dysfunctions but, a more comprehensive approach should be employed (not only EM): assessment of metabolism + measurement of mtDNA damage + measurement of oxidized cardiolipin for example.

Response: We truly appreciate your professional advice. According to your suggestions, we applied another method to better measure mitochondrial dysfunctions. An ATP (Adenosine triphosphate) assay kit was used to assess mitochondrial metabolism. Hope such modifications will satisfy you. Please help us check it (Fig. 3G and Fig. 4L).

11. Additionally, p9, the sentence “TEM images showed a greater incidence of mitochondrial damage” is an overstatement.

Response: We are sorry for the inappropriate description, and we have reversed this sentence on page 11, lines 312-316.

TEM images exhibited the mitochondrial damage including mitochondrial vacuolation, shrinkage, rupture, and fewer mitophagosomes in the si-Parkin group, which indicated that Parkin inhibition largely abolished the effects of ML.

12. The authors should use quantitative approach (such as flow cytometry) to assess mitochondrial density, MMP and mROS in addition to microscopy. With these regards, the methodology of quantification of IF images should be better explained. More generally, it is not clear how were the stats made for microscopy approaches.

Response: Thank you for your comment. We couldn't agree with you more. Flow cytometry is indeed more accurate way to assess these indicators. However, according to the manufacturer's instructions, the ratio of green to red fluorescence is dependent only on the membrane potential and not on other factors such as mitochondrial size, shape, and density that may influence single-component fluorescence signals. Use of fluorescence ratio detection therefore allows researchers to make comparative measurements of membrane potential and determine the percentage of mitochondria within a population that respond to an applied stimulus. In addition, many studies also used fluorescence microscopy to measure mitophagy-related indicators^{7,20,21}. We have added these referenced in the methods section on page 5, line 123. Please allow us not to use flow cytometry here.

13. The authors should use dedicated technology such as mito-Keima to quantify mitophagy

Response: Thank you for your kind suggestion. Mito-Keima is a pH-sensitive fluorescent protein located in mitochondria that allows the detection of mitochondria movement from the cytosol to lysosomes. In accordance with your advice, we loaded Mito-Keima plasmid with lentivirus and transfected into HUVECs but the transfection efficiency was not satisfactory and the final observation results were not good. And we still do not know the reason. Therefore, we purchased a mitophagy detection kit that can also detect mitophagy effectively^{18,22}. The representative images of mitophagy in HUVECs were detected with the mitophagy detection kit, wherein red staining indicates mitophagy, green staining indicates lysosomes, and yellow staining indicates co-localization of mitophagy and lysosomes (Fig. 2E, F). Hope these modifications will meet your requirements.

14. There are numerous spelling mistakes in the manuscript. Additionally, wording is sometimes surprising or vague e.g p4 “overdose of anesthetic”, p9: “Parkin is essential for mitophagy on most occasions”, p10: “the four groups were not significant”.

Response: We apologize for making these mistakes. We have amended them and marked them in red on page 6-7, lines 182-183; page10, lines 291-292; page 11-12, lines 333-334.

The English in this document has been checked by at least two professional editors, both native speakers of English. Please check if the revised version meets the English presentation standard. For a certificate, please see the attachment.

15. Use colors rather than grayscale in the figures. The bar graphs are not always easy to distinguish.

Response: According to your suggestion, we changed all the figures into color ones. It does make the figures more aesthetic and easier to distinguish.

16. The order of the panels should be re-arranged to follow the flow of the text.

Response: Many thanks for your suggestions. We re-arranged the panels as much as possible in order of the text. These revisions do make our manuscript easier to read.

Reviewer #2

In their article entitled “Activating Parkin-dependent mitophagy alleviates oxidative stress and apoptosis in human umbilical vein endothelial cells and promotes random-pattern skin flaps survival”, Chen et al. use an in vitro model system to dissect the molecular mechanisms responsible for the improvement of random-pattern skin flap survival upon melatonin treatment. To model the ischemia/reperfusion injury that is thought to occur upon random-pattern skin flap surgery, they submit HUVEC to oxidative stress in vitro (TBHP treatment). They claim that, in these conditions, melatonin (ML) treatment alleviates cell death by triggering mitophagy. They observe an involvement of Parkin and of the AMPK/TFEB pathway. Finally, they conduct in vivo experiments to confirm their in vitro findings.

Although the novelty of the conclusions is not major (the effect of ML on skin flaps as well as the effect on ML on mitophagy have been described previously in reference #13 and 14 as the authors

themselves acknowledge), the study is well conducted and brings new light on the role of melatonin in the context of random-pattern skin flaps survival. There are, however, several points that need to be addressed before publication:

Response:

Dear Professor Dancourt,

Thank you for your detailed comments and giving us the opportunity to improve our manuscript. Hope our reversion will satisfy you.

1. The authors claim that autophagy is increased in TBHP+ML conditions as they observed varying levels of the autophagy markers LC3II and p62. However, an increase in LC3II could be due to an increase in autophagy induction or a decrease of autophagic degradation. The authors should use a lysosomal inhibitor such as Bafilomycin A1 to block autophagic flux and to study whether their LC3II increase really reflects an increase in autophagy activation. The same is true for the decreased levels of p62 upon ML treatment, which could reflect an increase in autophagy or a decrease in oxidative stress, which is known to regulate p62 at the transcriptional level (pmid:20452972). The use of Bafilomycin A1 treatment on the in vitro HUVEC system is crucial to conclude here.

Response: We couldn't agree more with you, according to your advice, we used Bafilomycin A1 to validate the induction of mitophagy. Please check it on page 9, lines 262-269 and Fig. 2G, H; Fig. S2A, B. For your convenience, we displayed the sentences from the manuscript below.

Bafilomycin A1 (an inhibitor of lysosomal function) was used to validate the induction of mitophagy flux. As displayed in Fig. 2 G and H, the difference in LC3-II and P62 expression levels demonstrated that mitophagy flux was significantly enhanced by ML. To further investigate the effect of ML on mitochondria, we used a MitoTracker dye to measure the number of mitochondria in HUVECs. The fluorescence images of MitoTracker

demonstrated that ML administration avoided TBHP-induced mitochondrial depletion, while bafilomycin A1 reversed this effect (Fig. S2 A, B).

2. As one of the main conclusions of the study is that increasing mitophagy promotes random-pattern skin flaps survival, the authors should try to increase mitophagy in their system other than by ML treatment in order to rule out that what they observe is not due to a more general antioxidant effect of ML. Mitophagy activators have recently been described (PMCID: PMC7665171 for example).

Response: Thank you very much for your professional comment. According to your suggestion, UMI-77, a potent mitophagy inducer, is a great positive control. Whereas, the delivering date of UMI-77 is too long, so we can only choose another one as the positive control. Urolithin A is one the best characterized mitophagy inducer so far²³. Therefore, we used urolithin A as a positive control. Figure 2 has been reversed and the results part has been rewritten and marked in red on page 8-9, lines 241-273. Please check it.

3. Although it does not impact the general understanding of the results presented, significant rewrites should be made to improve the fluidity of the text. Some sentences are not very precise and need to be put in context. For instance: “Moreover, ATP production analysis suggested that ML enhanced mitochondria-derived ATP and inhibited mitophagy and blocked ML’s effect on mitochondria energy production (Fig. 3C)” (line 244).

Response: Thank you for your careful review. The English in this document has been checked by at least two professional editors, both native speakers of English. Please see if the revised version met the English presentation standard. For a certificate, please see the attachment.

We have reversed this sentence on page 10, lines 282-284. For your convenience, we displayed this sentence from the manuscript below.

For the assessment of mitochondrial metabolism, ATP production analysis suggested that ML

enhanced mitochondria-derived ATP, while CsA blocked ML's effect on mitochondria energy production (Fig. 3G)

4. line 234: Fig.2F does not exist. Co-localization studies should be quantified as they don't appear striking on the images alone (this is also true for FigS2).

Response: We apologize for such mistakes in figure numbers. Now figure 2 has been reversed and the results parts have been rewritten and marked in red on page 8-9, lines 241-273. Please check it. Moreover, we have quantified the Co-localization and showed by the histogram in Fig. 2C, D and Fig. S5C, D. Hope these modifications will meet your requirements.

5. The RNA-seq results are not well presented: where are the genes in Fig.4B on the Fig.4A plot? Presenting the sequencing data in a table should also be considered.

Response: We truly appreciate your careful comment. We reversed Fig. 4 and pointed out the corresponding genes at Fig.4A plot.

6. Please explain the effect of si-Control on the phenotypes studied.

Response: Thank you. As a negative control, si-Control only contains transfection agent. The transfection reagents were provided for from RiboBio Co., Ltd (Guangzhou, China), and no effect on experimental results was reported in previous studies^{24,25}. From our western blot results, that si-Control did not significantly change the expression of Parkin (Fig. 4C, D). In addition, we used another siRNA against Parkin to confirm these results. Hope this explanation will satisfy you.

7. Please show the efficiency of Nuclear/cytoplasmic fractionation (Figure 5).

Response: Many thanks. We have added the expression level of GAPDH in nuclear fraction

and H3 in cytoplasmic fraction to show the efficiency of nuclear/cytoplasmic extraction.
Please kindly check it in Fig 5A, E.

8. Several errors could be spotted in the text:

- line 227: Fig.2E, F should be replaced by Fig.2C, D
- line 406: “Founding” should be replaced by “Funding”
- line 572: “treated as above” should be elaborated.

Response: We are sorry for such mistakes.

1. Now figure 2 has been reversed and the results part has been rewritten and marked in red on page 8-9, lines 241-273.
2. The spelling mistake has been corrected on page 16, line 465.
3. We have rewritten the figure legend and elaborated the treatment.

9. Several errors could be spotted in the figures:

- FigS1B: μm should be replaced by μM
- Fig3A: western blots should be labeled PINK1 and not PINK
- Fig3G: TBHP+ML+CC should be replaced with TBHP+ML+CsA
- Fig3F: red and green panels were inverted (left panels)
- Fig3: panels should be re-ordered as to show 3C as the last panel of the figure
- text in Fig4A and Fig4B is way too small (see minor comment #3)
- Fig4I: replace “enlarged” with “higher magnification”
- Fig5C: does not seem to be the same format as the other panels. “TFEB” should be replaced with “TBHP”
- Fig7A: please show insets as labeling is difficult to see
- Fig8: explain the red arrow (should it be inhibitory?)

Response: Thank you for such detailed comments. We have revised the above content according to your opinion.

- FigS1B: μm has been replaced by μM .
- Fig3A: The wrong labeled has been corrected.

- Fig3G: We have replaced TBHP+ML+CC with TBHP+ML+CsA.
- Fig3: The order of the panel has been re-arranged.
- Fig4: We reversed Fig. 4 and pointed out the corresponding genes at Fig.4A plot followed by your suggestion. At the same time, we amended the font in the image to make it easier to read. Finally, we replace “enlarged” with “higher magnification”.
- Fig5C: Sorry, we made a mistake here. We have revised the wrong label.
- Fig7A: We adjust the scale bar to make it easy to see.
- Fig8: The red arrow represents Inhibition. We added annotation in figure 8.

Reviewer #3

The study of Zhengtai Chen and co-workers aimed at investigating the mechanism of mitophagy induced by Melatonin and its effect on the survival of skin flaps. From their results, the authors concluded that Melatonin induced mitophagy in human umbilical vein endothelial cells through a mechanism that involves Parkin 1. The induction of mitophagy alleviates oxidative stress and apoptosis. In in vivo study, the authors found that melatonin promotes mitophagy to enable flap survival. Overall, the manuscript is well-written and the topic is original. However, the authors have overinterpreted some data and additional studies are needed to reinforce the data of the manuscript.

Below are my specific concerns on the manuscript:

Response: Thank you very much for your professional and detail comments. We cherish this opportunity to revise our manuscript. We have made corresponding modifications according to your suggestions, and these modifications have significantly improved the quality of our manuscript.

1. My major concern is related to the induction of mitophagy by Melatonin. Figure 2A: a significant accumulation of LC3-II was observed in cells treated with TBHP+Melatonin versus TBHP alone but it is unclear if this accumulation is related to the induction or

inhibition of (mito)autophagy. If mitophagy is functional, the expression of LC3-II should decrease as LC3-II is degraded along this process. To validate the induction of mitophagy, experiment presented in the figure 2A should be repeated in the presence and absence of Bafilomycin A1 (an inhibitor of lysosomal function).

Response: We couldn't agree more with you, according to your advice, we used Bafilomycin A1 to validate the induction of mitophagy. Please check it on page 9, lines 262-269 and Fig. 2G, H; Fig. S2A, B. For your convenience, we displayed the sentences from the manuscript below.

Bafilomycin A1 (an inhibitor of lysosomal function) was used to validate the induction of mitophagy flux. As displayed in Fig. 2 G and H, the difference in LC3-II and P62 expression levels demonstrated that mitophagy flux was significantly enhanced by ML. To further investigate the effect of ML on mitochondria, we used a MitoTracker dye to measure the number of mitochondria in HUVECs. The fluorescence images of MitoTracker demonstrated that ML administration avoided TBHP-induced mitochondrial depletion, while bafilomycin A1 reversed this effect (Fig. S2 A, B).

2. Otherwise, I suggest to use the Mito-keima tandem plasmid which is a pH sensitive fluorescent protein located in mitochondria that allows the detection of mitochondria movement from cytosol to lysosomes.

Response: Thank you for your kind suggestion. Mito-Keima is a pH-sensitive fluorescent protein located in mitochondria that allows the detection of mitochondria movement from the cytosol to lysosomes. In accordance with your advice, we loaded Mito-Keima plasmid with lentivirus and transfected into HUVECs but the transfection efficiency was not satisfactory and the final observation results were not good. And we still do not know the reason. Therefore, we purchased a mitophagy detection kit that can also detect mitophagy^{18,22}. The representative images of mitophagy in HUVECs were detected with the mitophagy detection kit, wherein red staining indicates mitophagy, green staining

indicates lysosomes, and yellow staining indicates co-localization of mitophagy and lysosomes (Fig. 2E, F). Hope these modifications will meet your requirements.

3. **Figure 2C: To validate mitophagy, the number of mitochondria should be counted after staining cells with mitoTracker dye in the presence and absence of Bafilomycin A.**

Response: Thank you for your comment. According to your advice, we measured the number of mitochondria in the presence and absence of Bafilomycin A1. Please check it in Fig. S2A, B.

4. **Figure 2E: A better resolution of images is needed to appreciate the co-localization of LC3 and TOM20 in cells treated with TBHP+ML. Please note that LC3 and TOM20 staining were both increased under TBHP + ML. Can the authors provide an explanation for this observation?**

Response: In fact, the fluorescence of TOM20 did not increase in the TBHP + ML group. The increased fluorescence of TOM20 may be due to the difference in cell level at the time of photographing. We apologize for making such mistake. Please kindly allow us to reversed it and replace the picture with a higher resolution. The order of the panel has been re-arranged please check it in Fig. 2C, D.

5. **II) Figure 3A: the authors used CsA as a mitophagy inhibitor and found that inhibition of autophagy led to a decrease of both LC3-II and p62 expression levels. This is puzzling as normally these proteins are accumulated during inhibition of mitophagy. Can the authors provide an explanation for this observation?**

Response: Sorry to make you confused. It might be our inaccuracy expression to let you

misunderstood. Actually, in our study, the level of p62 was increased and the protein level of LC3-II, Parkin, PINK1 was downregulated in TBHP + ML + CsA group compared with TBHP + ML group, which indicated that CsA successfully inhibited ML induced mitophagy. In addition, in some previous studies, they also found the decreased level of LC3-II and the accumulated level of P62 after the use of CsA^{18,19}. Hope our explanation will satisfy you.

6. **Figure 3F: quantification is needed for MPP.**

Response: Thank you very much. The quantification of MPP was demonstrated by a histogram in figure 3E.

7. **Figure 4: To support the author's conclusion that Parkin 1 is involved in mitophagy, the authors should determine the effect of si-Parkin 1 on the mitochondria number (through staining with mitoTracker) and the expression levels of LC3-II and p62, as well. Moreover, it would be nice to have two siRNAs against Parkin 1 to confirm the results shown in Figure 4.**

Response: Thank you very much for your suggestion.

1. According to your advice, we applied another siRNAs against Parkin 1 to better confirm our results.
2. The mitochondria number were measured by mitoTracker in Fig. S4A, B.
3. The expression levels of LC3-II and p62 were measured by western blot in Fig. 4E, F.

8. **Figure 5: Figure 5A: please add the expression level of GAPDH in nuclear fraction and vis versa of H3 in cytoplasmic fraction (to show the efficiency of nuclear/cytoplasmic extraction). Moreover, it would be nice to confirm the subcellular localization of TFEB by immunofluorescence study in the experiments presented in figures 5A and 5B.**

Response: Thank you. We adjusted figure 5 and added the expression level of GAPDH in

nuclear fraction and the expression level of H3 in cytoplasmic fraction in Fig. 5A, B and Fig. 5E, F. Moreover, the subcellular localization of TFEB was also be measured by immunofluorescence of TFEB (Fig. S5A, B). These modifications indeed made our data stronger to support the statement our manuscript.

9. III) The authors should avoid over-interpretation of their “TFEB” results. The authors stated that “TFEB was indeed involved in the regulation of mitophagy by ML” (Line 383) but there is no evidence in the paper for such regulation. To support this statement, I suggest to examine the effect of TFEB si/shRNA on mitophagy-induced by Melatonin. Otherwise, the authors should rewrite the discussion part and change the figure 8 in accordance with their results.

Response: We truly appreciate your professional comments. Followed by your comments we transfected a TFEB si-RNA to better verify the TFEB’s role in ML-induced mitophagy. Then we measured the mitophagy flux in the presence and absence of Bafilomycin A1. As shown in Fig. S6C and D, TFEB knockdown significantly blocked the ML-induced mitophagy flux. Additionally, the decreased C-caspase3 and SOD1 expression level in Fig. S6E and F, reflected the anti-apoptosis and anti-oxidative effects of ML were antagonized by TFEB knockdown. Your professional suggestions do perfect the logic of our manuscript.

10. - A better resolution of images is needed for Fig 2A (c-caspase 3) and Fig 3A (p62).
- Figure 5 C: the authors mistakenly wrote TFEB instead of TBHP.

Response: Many thanks for your careful check.

- According to you suggestion, we have replaced Fig 2A (c-caspase 3) and Fig 3A (p62) with a higher resolution of images. Now the order of the panel has been re-arranged please check it in Fig. S2C and Fig 3A.
- Figure 5 C: Sorry, we made a mistake here. We have revised the wrong label.

11. In my opinion, the data are not strong enough to warrant the conclusions reached by the authors. This study requires further evidence to support the induction of mitophagy by melatonin and its regulation.

Response: These comments are valuable and very helpful for improving our manuscript. Our studies do have some shortcomings. But now with your modification, the quality of the manuscript has improved. We have studied comments carefully and made corrections which we hope to meet with your approval.

- 1 van den Heuvel, M., Buurman, W., Bast, A. & van der Hulst, R. Review: Ischaemia-reperfusion injury in flap surgery. *J. Journal of plastic, reconstructive aesthetic surgery*. **62**, 721-726 (2009). doi: 10.1016/j.bjps.2009.01.060.
- 2 Balaban, R., Nemoto, S. & Finkel, T. Mitochondria, oxidants, and aging. *J. Cell*. **120**, 483-495 (2005). doi: 10.1016/j.cell.2005.02.001.
- 3 West, A., Brodsky, I., Rahner, C., Woo, D., Erdjument-Bromage, H., Tempst, P., . . . Ghosh, S. TLR signalling augments macrophage bactericidal activity through mitochondrial ROS. *J. Nature*. **472**, 476-480 (2011). doi: 10.1038/nature09973.
- 4 Green, D. & Van Houten, B. SnapShot: Mitochondrial quality control. *J. Cell*. **147**, 950, 950.e951 (2011). doi: 10.1016/j.cell.2011.10.036.
- 5 Gu, C., Li, L., Huang, Y., Qian, D., Liu, W., Zhang, C., . . . Yin, G. Salidroside

Ameliorates Mitochondria-Dependent Neuronal Apoptosis after Spinal Cord Ischemia-Reperfusion Injury Partially through Inhibiting Oxidative Stress and Promoting Mitophagy. *J. Oxidative medicine cellular longevity*. **2020**, 3549704 (2020). doi: 10.1155/2020/3549704.

6 Wang, J., Zhu, P., Li, R., Ren, J. & Zhou, H. Fundc1-dependent mitophagy is obligatory to ischemic preconditioning-conferred renoprotection in ischemic AKI via suppression of Drp1-mediated mitochondrial fission. *J. Redox biology*. **30**, 101415 (2020). doi: 10.1016/j.redox.2019.101415.

7 Cao, S., Wang, C., Yan, J., Li, X., Wen, J., Hu, C. & medicine. Curcumin ameliorates oxidative stress-induced intestinal barrier injury and mitochondrial damage by promoting Parkin dependent mitophagy through AMPK-TFEB signal pathway. *J. Free radical biology medicine*. **147**, 8-22 (2020). doi: 10.1016/j.freeradbiomed.2019.12.004.

8 Gbahou, F., Cecon, E., Viault, G., Gerbier, R., Jean-Alphonse, F., Karamitri, A., . . . Jockers, R. Design and validation of the first cell-impermeant melatonin receptor agonist. *J. British journal of pharmacology*. **174**, 2409-2421 (2017). doi: 10.1111/bph.13856.

9 Kerem, H., Akdemir, O., Ates, U., Uyanıkgıl, Y., Demirel Sezer, E., Bilkay, U., . . . Songur, E. The effect of melatonin on a dorsal skin flap model. *J. Journal of investigative surgery*. **27**, 57-64 (2014). doi: 10.3109/08941939.2013.835892.

10 Ali Gurlek, H. A., Hakan Parlakpınar, Aysun Bay-Karabulut, & Mehmet Celik, N. S. a. A. Protective effect of melatonin on random pattern skin flap necrosis in pinealectomized rat. *J. Pineal Research*. **36**, 58–63 (2004).

- 11 Tunç, S., Kesiktaş, E., Yılmaz, Y., Açikalin, A., Oran, G., Yavuz, M., . . . Eser, C. Assessing the effects of melatonin and N-acetylcysteine on the McFarlane flap using a rat model. *J. Plastic surgery*. **24**, 204-208 (2016). doi: 10.1177/229255031602400302.
- 12 Wu, H., Chen, H., Zheng, Z., Li, J., Ding, J., Huang, Z., . . . disease. Trehalose promotes the survival of random-pattern skin flaps by TFEB mediated autophagy enhancement. *J. Cell Death & Disease*. **10**, 483 (2019). doi: 10.1038/s41419-019-1704-0.
- 13 Kira, T., Omokawa, S., Akahane, M., Shimizu, T., Nakano, K., Nakanishi, Y., . . . Tanaka, Y. Effectiveness of Bone Marrow Stromal Cell Sheets in Maintaining Random-Pattern Skin Flaps in an Experimental Animal Model. *J. Plastic reconstructive surgery*. **136**, 624e-632e (2015). doi: 10.1097/prs.0000000000001679.
- 14 Lou, Z., Zhang, C., Li, J., Chen, R., Wu, W., Hu, X., . . . Zhao, Q. β Apelin/APJ-Manipulated CaMKK/AMPK/GSK3 Signaling Works as an Endogenous Counterinjury Mechanism in Promoting the Vitality of Random-Pattern Skin Flaps. *J. Oxidative medicine cellular longevity*. **2021**, 8836058 (2021). doi: 10.1155/2021/8836058.
- 15 Geng, L., Zhang, G., Yao, M. & Fang, Y. Rip 1-dependent endothelial necroptosis participates in ischemia-reperfusion injury of mouse flap. *J. Journal of dermatological science*. **97**, 30-40 (2020). doi: 10.1016/j.jdermsci.2019.11.009.
- 16 Zhou, F., Zhang, L., Chen, L., Xu, Y., Chen, Y., Li, Z., . . . Qi, S. Prevascularized mesenchymal stem cell-sheets increase survival of random skin flaps in a nude mouse model. *J. American journal of translational research*. **11**, 1403-1416 (2019).

- 17 Henderson, P., Singh, S., Belkin, D., Nagineni, V., Weinstein, A., Weissich, J. & Spector, J. Hydrogen sulfide protects against ischemia-reperfusion injury in an in vitro model of cutaneous tissue transplantation. *J. The Journal of surgical research.* **159**, 451-455 (2010). doi: 10.1016/j.jss.2009.05.010.
- 18 Lin, J., Zhuge, J., Zheng, X., Wu, Y., Zhang, Z., Xu, T., . . . Wang, X. Urolithin A-induced mitophagy suppresses apoptosis and attenuates intervertebral disc degeneration via the AMPK signaling pathway. *J. Free radical biology medicine.* **150**, 109-119 (2020). doi: 10.1016/j.freeradbiomed.2020.02.024.
- 19 Chen, L., Shi, X., Xie, J., Weng, S., Xie, Z., Tang, J., . . . Yang, L. Apelin-13 induces mitophagy in bone marrow mesenchymal stem cells to suppress intracellular oxidative stress and ameliorate osteoporosis by activation of AMPK signaling pathway. *J. Free radical biology medicine.* **163**, 356-368 (2021). doi: 10.1016/j.freeradbiomed.2020.12.235.
- 20 Dolman, N., Chambers, K., Mandavilli, B., Batchelor, R. & Janes, M. Tools and techniques to measure mitophagy using fluorescence microscopy. *J. Autophagy.* **9**, 1653-1662 (2013). doi: 10.4161/auto.24001.
- 21 Wang, B., Yin, X., Gan, W., Pan, F., Li, S., Xiang, Z., . . . Li, D. PRCC-TFE3 fusion-mediated PRKN/parkin-dependent mitophagy promotes cell survival and proliferation in PRCC-TFE3 translocation renal cell carcinoma. *J. Autophagy.* 1-19 (2020). doi: 10.1080/15548627.2020.1831815.
- 22 Iwashita, H., Torii, S., Nagahora, N., Ishiyama, M., Shioji, K., Sasamoto, K., . . . Okuma, K. Live Cell Imaging of Mitochondrial Autophagy with a Novel Fluorescent

- Small Molecule. *J. ACS chemical biology*. **12**, 2546-2551 (2017). doi: 10.1021/acscchembio.7b00647.
- 23 Ryu, D., Mouchiroud, L., Andreux, P., Katsyuba, E., Moullan, N., Nicolet-Dit-Félix, A., . . . Auwerx. Urolithin A induces mitophagy and prolongs lifespan in *C. elegans* and increases muscle function in rodents. *J. Nature medicine*. **22**, 879-888 (2016). doi: 10.1038/nm.4132.
- 24 Xu, D., Jin, H., Wen, J., Chen, J., Chen, D., Cai, N., . . . Wang, X. Hydrogen sulfide protects against endoplasmic reticulum stress and mitochondrial injury in nucleus pulposus cells and ameliorates intervertebral disc degeneration. *J. Pharmacological research*. **117**, 357-369 (2017). doi: 10.1016/j.phrs.2017.01.005.
- 25 Zhang, Z., Xu, T., Chen, J., Shao, Z., Wang, K., Yan, Y., . . . Wang, X. Parkin-mediated mitophagy as a potential therapeutic target for intervertebral disc degeneration. *J. Cell death disease*. **9**, 980 (2018). doi: 10.1038/s41419-018-1024-9.

Reviewers' comments:

Reviewer #1 (Remarks to the Author):

I am satisfied with the answers provided by the authors and the corresponding revised version of the manuscript. The new data significantly improved the quality of this interesting work.

Reviewer #2 (Remarks to the Author):

In the revised version of their article entitled « Activating Parkin-dependent mitophagy alleviated oxidative stress, apoptosis and promoted random-pattern skin flaps survival”, Chen et al. made meaningful changes that, in my opinion, bring the study to publication standards.

I would however, suggest that the title not be written in the past tense but in the present tense. I therefore recommend the publication of this manuscript in Communications Biology.

Reviewer #3 (Remarks to the Author):

The authors have made several modifications regarding the original requests of reviewers.

Although, this manuscript has been improved, several questions remain unanswered.

In particular, it remains unclear if ML promotes mitophagy and TFEB induction in umbilical vein endothelial cells.

In my original review, I requested that the authors provide additional evidence for the induction of mitophagy by ML. The new results obtained by Mitotracker, showed that ML promotes an increase in the number of mitochondria and the addition of Bafilomycin A1 to ML prevents this response ((figure 2 A,B). This is puzzling as the addition of Bafilomycin A1 in the context of mitophagy should normally prevent the degradation of mitochondria and thus increase the number of mitochondria. Moreover, if mitophagy is mediated by Parkin 1 normally Parkin1 knockdown should increase the number of the mitochondria but the authors found opposite results (figure S4). This is also puzzling and do not argue in favor of the induction of mitophagy by ML.

In conclusion, the authors misinterpreted their results and the data presented in figures 2 A,B; fig S4) do not match with the authors conclusion that ML induces mitophagy.

In my previous review, I requested that the authors provide additional evidence for the role of TFEB in ML-induced mitophagy.

Although, the authors included additional data, the results do not argue in favor of ML-induced mitophagy via TFEB activation.

- Figure S5 A showed the effect of TBHP but not ML on TFEB nuclear translocation. In this figure, we can notice that control cells display nuclear translocation of TFEB and TBHP promotes cytoplasmic localization of TFEB. Could the authors explain these data?

- From data presented in the figure 5, the authors concluded that ML-induced mitophagy through TFEB but there is no evidence in this figure for such regulation.

Reviewer #2

In the revised version of their article entitled « Activating Parkin-dependent mitophagy alleviated oxidative stress, apoptosis and promoted random-pattern skin flaps survival», Chen et al. made meaningful changes that, in my opinion, bring the study to publication standards.

I would, however, suggest that the title not be written in the past tense but the present tense.

I, therefore, recommend the publication of this manuscript in *Communications Biology*.

Response: We truly appreciate your professional advice. The title has been written in the present tense now.

The revised title is displayed below for your convenience.

Activating Parkin-dependent mitophagy alleviates oxidative stress, apoptosis, and promotes random-pattern skin flaps survival.

Reviewer #3 :

The authors have made several modifications regarding the original requests of reviewers. Although, this manuscript has been improved, several questions remain unanswered.

In particular, it remains unclear if ML promotes mitophagy and TFEB induction in umbilical vein endothelial cells.

1) In my original review, I requested that the authors provide additional evidence for the induction of mitophagy by ML. The new results obtained by Mitotracker, showed that ML promotes an increase in the number of mitochondria and the addition of Bafilomycin A1 to ML prevents this response ((figure 2 A, B). This is puzzling as the addition of Bafilomycin A1 in the context of mitophagy should normally prevent the degradation of mitochondria and thus increase the number of mitochondria. Moreover, if mitophagy is mediated by Parkin 1 normally Parkin1 knockdown should increase the number of the mitochondria but the authors found opposite results (figure S4). This is also puzzling and do not argue in favor of the induction of mitophagy by ML.

Response: We apologize for making you confused. It is because of the Mitotracker fluorescence intensity are susceptible to potential oxidases and Mitotracker fluorescence is related to the level of mitochondrial membrane potential¹⁻⁴. Tang et al. reported that trehalose promoted mitophagy and increased Mitotracker fluorescence intensity, but that this effect was abolished when mitophagy flow was blocked⁵. Padman et al. observed that inhibiting mitophagy reduced the fluorescence intensity Mitotracker in both yeast and human cells⁶. These studies share the same results of our study. As a result, the fluorescence intensity Mitotracker results in our study may not reflect accurate mitochondrial numbers.

Actually, in Fig. 4J our transmission electron microscopy images can directly exhibit the number of mitochondria in the si-Parkin group and is considerably higher than that in the si-Con group. However, the mitochondrial in the si-Parkin group exhibited mitochondrial vacuolation, shrinkage, rupture, and fewer mitophagosomes in the si-Parkin group, which indicated that Parkin inhibition largely abolished the effects of ML.

According to your advice we repeated the experiment and applied transmission electron microscopy to observe it. As exhibited in Fig. S2 C, the number of mitochondrial in ML + TBHP group was less than THBP group but the morphology of mitochondrial are more complete, including clearer mitochondrial crest, intact mitochondrial membrane, and more autophagolysosomes formation. However, when treated with BafA1 blocked these effects of ML. The THBP + ML + BafA1 group demonstrated mitochondrial vacuolation, shrinkage, and mitochondrial crest rupture. In summary, we can draw a conclusion that that ML upregulated mitophagy in HUVECs. Moreover, we have discussed the difference between the Mitotracker and the transmission electron microscopy results and marked in red on page15 line433-442. Hope these modifications will meet your requirements.

2) In my previous review, I requested that the authors provide additional evidence for the role of TFEB in ML-induced mitophagy.

Although, the authors included additional data, the results do not argue in favor of ML-induced mitophagy via TFEB activation.

- Figure S5 A showed the effect of TBHP but not ML on TFEB nuclear translocation. In this figure, we can notice that control cells display nuclear translocation of TFEB and TBHP promotes cytoplasmic localization of TFEB. Could the authors explain these data?

Response:

We apologize for the insufficient data about the ML on TFEB nuclear translocation. Therefore, we repeated the immunofluorescence of TFEB (Fig 5C and D). Since the control group did not receive any stimulations so it was unable to depict the TFEB nuclear translocation. ML, on the other hand, boosted TFEB nuclear translocation as compared to the control group. Furthermore, THBP did not promote the TFEB cytoplasmic localization but did reduce the nuclear transcription of TFEB. There was no significant difference in TFEB content in the cytoplasm across the three groups, which also matched our western bolt results. We hope these adjustments satisfy your expectations.

3) From the data presented in figure 5, the authors concluded that ML-induced mitophagy through TFEB but there is no evidence in this figure for such regulation.

Response: Thank you for your careful check about our manuscript. Your confusion may have been caused by our separate placement of supplementary figure 6 data. We re-arranged the panels and integrated supplementary figures 5 and 6 into the main figures. We revised this part and marked it in red on page11-12, lines 322-337. Indeed, this study about TFEB is not enough, and we will further strengthen it in our following study. Therefore, this is a limitation of our study and we tempered our conclusions and marked it in red on page16, line471-473. Please kindly check it.

The revised part is displayed below for your convenience.

TFEB is known for its ability to regulate lysosomal biogenesis and autophagy^{7,8}. Whether TFEB participates in ML induced mitophagy is unknown. According to the RNA sequencing results, the TFEB gene was downregulated by TBHP treatment and upregulated after ML treatment, signifying that TFEB participated in ML-induced mitophagy (Fig. 4B). As seen in the western blot results, TBHP suppressed TFEB nuclear expression whereas ML increased it, as compared to the control group. It's worth mentioning that the cytoplasmic TFEB expression level was somewhat higher in the ML group, although this was not statistically significant. Similarly, TFEB immunofluorescence staining validated the western blot results that ML promotes the nuclear translocation of TFEB (Fig. 5C, D). To confirm the role of TFEB in ML-induced mitophagy, a TFEB si-RNA was used. The western blot revealed that si-TFEB effectively suppressed TFEB (Fig. 5E, F). Then the autophagy-related proteins LC3 and P62 were next measured. In si-TFEB group, the decreased expression of LC3 and increased expression of P62, indicating that ML-induced autophagy was inhibited. Additionally, the upregulated C-caspase3 and downregulated SOD1 expression levels, reflected the anti-apoptotic and anti-oxidative effects of ML were antagonized by TFEB knockdown (Fig. 5H, I). All in all, it could be speculated that ML induced mitophagy in HUVECs may through upregulating the nuclear translocation of TFEB.

Reference

- 1 Zhang, R. et al. Reaction-free and MMP-independent fluorescent probes for long-term mitochondria visualization and tracking. *Chemical science* 10, 1994-2000, doi:10.1039/c8sc05119d (2019).
- 2 Rodriguez-Enriquez, S., Kim, I., Currin, R. & Lemasters, J. Tracker dyes to probe mitochondrial autophagy (mitophagy) in rat hepatocytes. *Autophagy* 2, 39-46, doi:10.4161/auto.2229 (2006).
- 3 Pokharel, S., Gliyazova, N., Dandepally, S., Williams, A. & Ibeanu, G. In vitro Neuroprotective effects of an BBB permeable phenoxythiophene sulfonamide small molecule in glutamate-induced oxidative injury. *Experimental therapeutic medicine* 23, 79, doi:10.3892/etm.2021.11002 (2022).
- 4 Clutton, G., Mollan, K., Hudgens, M. & Goonetilleke, N. A Reproducible, Objective Method Using MitoTracker® Fluorescent Dyes to Assess Mitochondrial Mass in T Cells by Flow Cytometry. *the journal of the International Society for Analytical Cytology* 95, 450-456, doi:10.1002/cyto.a.23705 (2019).
- 5 Tang, Q. et al. Trehalose ameliorates oxidative stress-mediated mitochondrial dysfunction and ER stress via selective autophagy stimulation and autophagic flux restoration in osteoarthritis development. *Cell death disease* 8, e3081, doi:10.1038/cddis.2017.453 (2017).
- 6 Padman, B., Bach, M., Lucarelli, G., Prescott, M. & Ramm, G. The protonophore CCCP interferes with lysosomal degradation of autophagic cargo in yeast and mammalian cells. *Autophagy* 9, 1862-1875, doi:10.4161/auto.26557 (2013).
- 7 Lu, X. et al. AMPK protects against alcohol-induced liver injury through UQCRC2 to up-regulate mitophagy. *Autophagy*, 1-22, doi:10.1080/15548627.2021.1886829 (2021).
- 8 Sass, F. et al. TFEB deficiency attenuates mitochondrial degradation upon brown adipose tissue whitening at thermoneutrality. *Molecular metabolism* 47, 101173, doi:10.1016/j.molmet.2021.101173 (2021).

Reviewer #1:

I am satisfied with the answers provided by the authors and the corresponding revised version of the manuscript. The new data significantly improved the quality of this interesting work.

Response: Thank you very much for your professional and thoughtful comments.